# Conserved and species-specific chromatin remodeling and regulatory dynamics during mouse and chicken limb bud development

Shalu Jhanwar [1,2✉], Jonas Malkmus[1], Jens Stolte[1,3], Olga Romashkina[1], Aimée Zuniga [1✉] & Rolf Zeller [1✉]

Chromatin remodeling and genomic alterations impact spatio-temporal regulation of gene expression, which is central to embryonic development. The analysis of mouse and chicken limb development provides important insights into the morphoregulatory mechanisms, however little is known about the regulatory differences underlying their morphological divergence. Here, we identify the underlying shared and species-specific epigenomic and genomic variations. In mouse forelimb buds, we observe striking synchrony between the temporal dynamics of chromatin accessibility and gene expression, while their divergence in chicken wing buds uncovers species-specific regulatory heterochrony. In silico mapping of transcription factor binding sites and computational footprinting establishes the developmental time-restricted transcription factor-DNA interactions. Finally, the construction of target gene networks for HAND2 and GLI3 transcriptional regulators reveals both conserved and species-specific interactions. Our analysis reveals the impact of genome evolution on the regulatory interactions orchestrating vertebrate limb bud morphogenesis and provides a molecular framework for comparative Evo-Devo studies.

[1] Developmental Genetics, Department of Biomedicine, University of Basel, CH-4058 Basel, Switzerland. [2] Swiss Institute for Bioinformatics, University of Basel, CH-4058 Basel, Switzerland. [3] Present address: Nestlé Institute of Health Sciences, EPFL Innovation Park, CH-1015 Lausanne, Switzerland. ✉email: shalu.jhanwar@unibas.ch; aimee.zuniga@unibas.ch; rolf.zeller@unibas.ch

Precise spatiotemporal regulation of gene expression is the hallmark of normal embryonic development. Transcription factor (TF) complexes regulate gene expression dynamics via interactions with specific *cis*-regulatory elements (CREs) that function as transcriptional enhancers or repressors. With the exception of a few TFs preferentially binding within heterochromatin, the vast majority of TFs interact within accessible chromatin[1]. Therefore, the spatiotemporal modulation of chromatin accessibility is essential to understand the *cis*-regulatory control of gene expression during cell fate specification and organ/tissue morphogenesis. Recent genome-wide studies have highlighted the importance of chromatin accessibility for gene regulation during embryonic development[2], cell fate determination[3], differentiation[4], and disease[5]. In particular, systematic profiling of chromatin accessibility and gene expression has uncovered the regulatory dynamics that orchestrate heart tube development in zebrafish embryos[2] and the gene regulatory interactions governing retinal morphogenesis[6]. Integrative analysis of human and bovine preimplantation development established that chromatin remodeling resulting in accessible chromatin is key to embryonic genome activation[7,8]. Finally, comparative analysis of mouse and pig limb buds identified variations in *cis*-regulatory landscapes that likely participate in the functional emergence of Artiodactyl limb traits[9,10]. Another mechanism contributing to functional divergence of gene regulation is the excessive number of base substitutions that accumulated in a species within otherwise conserved vertebrate CREs. Many of the accelerated regions[11] are developmental enhancers with divergent spatiotemporal activities. In a few cases, they have been correlated with species-specific changes in the expression of the associated target genes, which might contribute to shaping specific morphological traits. For instance, a bat accelerated region encoding a forelimb-specific enhancer in the HoxD cluster could function in the development of the highly derived bat forelimb autopod into a wing[12]. Similarly, human accelerated regions include enhancers whose activities differ from their chimpanzee orthologues, which have been proposed to function in the development of human-specific traits[11].

Mouse and chicken limb buds are experimental paradigms to study gene regulation and the signaling pathways controlling proliferation and survival, patterning and differentiation during vertebrate organogenesis[13–15]. Reverse genetics in the mouse has established the essential regulatory roles of key limb genes and their signals. Briefly, the nascent limb bud mesenchyme is polarized along its anteroposterior (AP) axis by mutually antagonistic interactions of the GLI3 repressor (GLI3R) and HAND2 transcriptional regulators. These TFs act upstream of SHH signaling and trigger the establishment of distinct anterior and posterior mesenchymal domains[16–18]. SHH signaling propagates *Fgf* expression in the apical ectodermal ridge (AER) via transcriptional upregulation of *Gremlin1* (GREM1)-mediated BMP antagonism in the distal limb bud mesenchyme. These feedback signaling interactions between mesenchyme and AER define the self-regulatory and self-terminating limb bud signaling system[14,19–21]. Given that conserved molecular pathways and interactions govern tetrapod limb development, mouse and chicken are two complementary models used to study the fundamental mechanisms underlying limb bud patterning and outgrowth. Recent studies have reported genomic locus-specific differences in the spatiotemporal expression of 5′ *Hoxa/d* and *Grem1* between mouse and chicken limb buds, thereby highlighting species-specific regulation[22–24]. However, the extent of their conserved and species-specific differences in the molecular control of early limb patterning and outgrowth has not been extensively studied.

In this study, we adopt a comprehensive approach integrating next-generation sequencing and genetics within a comparative framework. In particular, we leverage genome-wide comprehensive assessment of chromatin accessibility and transcriptional changes during mouse (*Mus musculus*) forelimb and chicken (*Gallus gallus*) wing bud development. Our analysis reveals the striking synchrony between temporal dynamics of chromatin accessibility and gene expression in mouse forelimb buds in contrast to stage-specific divergence in chicken wing buds. In addition, TF binding site (TFBS) enrichment and computational footprints of TFs uncover their developmental stage-specific interactions with DNA during mouse forelimb and chicken wing bud development. Integration of chromatin accessibility, putative TFBS, and gene expression allow the identification of candidate target genes for HAND2 and GLI3 in both species, which reveals both conserved and species-specific interactions. Finally, comparative sequence analysis of developmental enhancers reveals chicken accelerated regions (CARs) with activities divergent from their mouse orthologs in transgenic enhancer assays. Together, our study reveals the impact of genome evolution on the conserved and diversified gene regulatory trajectories underlying mouse forelimb and chicken wing bud development.

## Results

**Global gene expression dynamics in mouse forelimb buds reveal distinct temporal modules.** To study the gene expression dynamics during limb bud patterning and outgrowth, the transcriptome of mouse forelimb buds was profiled using RNA-seq at embryonic days E9.75, E10.5, and E11.5. Briefly, E9.75 overlaps with the onset of signaling interactions and outgrowth, E10.5 marks the developmental period controlled by the self-regulatory SHH/GREM1/AER-FGF signaling system, and E11.5 underlies the progression of the autopod development and self-termination of the morpho-regulatory signaling system (Fig. 1a)[19,20,25]. Principal components analysis (Fig. 1b, PC1-42% and PC2-17%) and hierarchical clustering (Supplementary Fig. 1a) of the gene expression profiles establish the low variability among biological replicates. Differentially expressed genes (DEGs) were identified (Fig. 1c, Supplementary Fig. 1b and Supplementary Dataset 1) by pairwise comparisons of developmental stages viz. E9.75/E10.5 ($n = 2157$), E10.5/E11.5 ($n = 1849$), and E9.75/E11.5 ($n = 3738$). Unbiased clustering of DEGs using *k-means* revealed six temporal modules characterized by distinct gene expression kinetics (Fig. 1d, Supplementary Fig. 1c, and Supplementary Dataset 1). Each DEG module consists of co-expressed genes that possess similar temporal gene expression trajectories during mouse forelimb bud development (Fig. 1e and Supplementary Fig. 1c) and standard silhouette analysis[26] underscores the robust partitioning of these modules (average silhouette coefficient: 0.51; Supplementary Fig. 1d). For follow-up analysis, the distinct temporal trajectory of each DEG module was assigned a specific identifier based on either its highest (hi) or lowest (lo) relative expression at a particular forelimb bud stage (Fig. 1d). For instance, the E9.75$^{hi}$ module encompasses DEGs with the highest expression at E9.75 relative to subsequent stages. Although genes belonging to E9.75$^{hi}$ and E11.5$^{lo}$ DEG modules show the highest expression at E9.75 and their expression decreases during limb bud outgrowth, these two modules are distinct due to their significantly different expression levels at the transition point (E10.5, top and bottom panels in Fig. 1d; box plots in Supplementary Fig. 1c). In particular, the decrease in gene expression between E9.75 and E10.5 is much larger for the E9.75$^{hi}$ than the E11.5$^{lo}$ DEG module. The converse is true for genes belonging to the

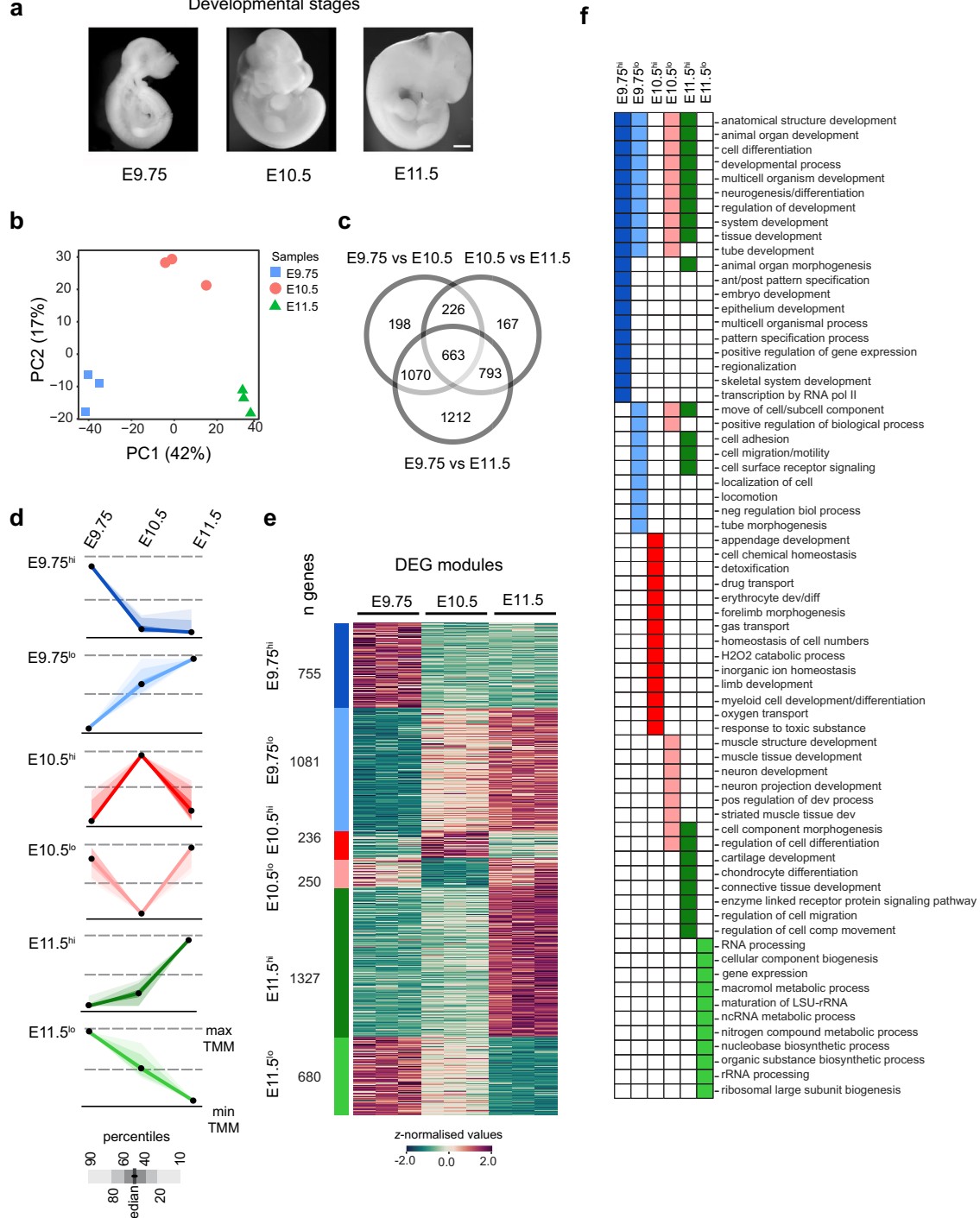

**Fig. 1 Transcriptome profiling of mouse forelimb bud identifies distinct temporal DEG modules. a** Illustrations of mouse embryos at embryonic days E9.75, E10.5, and E11.5 that were used for the analysis of forelimb bud development (scale bar: 250 µm). **b** Principal component analysis (PCA) of the RNA-seq samples of mouse forelimb buds (E9.75, E10.5, E11.5, $n = 3$ biological replicates per stage) were plotted as a function of the highly variable genes in PCA space. Genes were normalized using the Trimmed Mean of M-values (TMM) method of edgeR to carry out this analysis. The clustering of biological replicates ($n = 3$) at the different stages points to minimal intra-sample variability. Samples are indicated by different colors and shapes. **c** Three-way Venn diagram illustrates the overlap of DEGs across different pairwise comparisons. **d** Scaled module profiles were derived using *k-means* clustering of DEGs. The six DEG modules with distinct expression profiles are indicated by a color code that is used for all subsequent analyses. Black dots represent the median values while the shaded areas correspond to the percentiles as indicated in the scale (below the graphs). **e** Heatmap of all DEGs ($n = 4329$) illustrating relative gene expression across developmental stages. The DEGs were identified using the likelihood ratio test as implemented in glm function of edgeR. The adjustments were made for multiple comparisons using the Benjamini–Hochberg (BH) method. DEGs with significant changes in transcript levels must have a linear absolute FC cutoff of ≥1.5 and an adjusted $p$ value ≤ 0.05. The $z$-score scale represents mean-subtracted regularized log-transformed read counts. The number of DEGs per module ($n$) is indicated. **f** The top enriched GO terms for each DEG module are shown. DEG differentially expressed genes, GO gene ontology. Source data are provided as a Source Data file.

E9.75$^{lo}$ and E11.5$^{hi}$ DEG modules. Overall, all DEG modules display distinct expression kinetics in developing mouse forelimb buds. The majority of DEGs belong to the E11.5$^{hi}$ module (30.7%), followed by the E9.75$^{lo}$ module (25%), while the other DEG modules collectively account for 44.3% of all DEGs, Supplementary Fig. 1e).

Gene Ontology (GO) analysis of DEG modules revealed the enrichment of developmental and biological processes (Fig. 1f and Supplementary Dataset 2). In particular, the DEGs belonging to the E9.75$^{hi}$ module function predominantly in development and patterning, while the DEGs of the E11.5$^{lo}$ module are active in biogenesis-related processes. The DEGs of the E10.5$^{hi}$ and E10.5$^{lo}$ modules function preferentially in limb, muscle, neuronal development, and/or tissue homeostasis. By contrast, DEGs of the E11.5$^{hi}$ module show enrichment for developmental processes involving *Hox* family members and differentiation of chondrocytes, cartilage, and connective tissues (e.g. *Col* and *Runx*)[27]. This is in agreement with the ongoing chondrogenic differentiation of skeletal primordia in the proximal forelimb bud mesenchyme at E11.5, while 5'*Hoxa/d* genes function in the distal mesenchyme during autopod development[28]. In summary, the DEG modules (Fig. 1d–f) capture the molecular transitions underlying the progression of mouse forelimb bud development.

**Chromatin remodeling is associated with differential gene expression during mouse forelimb bud development.** Given the crucial role of the dynamically established chromatin accessibility in controlling gene expression[2,8,29,30], we investigated chromatin remodeling in mouse forelimb buds using ATAC-seq[31] at three developmental stages (Fig. 2). The high quality of the ATAC-seq datasets was verified by assessing nucleosome periodicity, the strong correlation between replicates (Supplementary Fig. 2a, b), and the quality of Tn5 transposase insertions at CTCF binding regions (Fig. 2a)[31]. In agreement with the previous studies[1], accessible chromatin regions are preferentially located in distal intergenic, intronic regions, and at promoters (≤1 kb from the nearest transcriptional start site TSS, Supplementary Fig. 2c, d). Further, the majority of accessible regions (>70%) overlaps with at least one histone mark indicative of active chromatin (Supplementary Fig. 2e), thereby supporting their potential functions as CREs.

To gain insight into the chromatin accessibility dynamics in mouse forelimb buds, we assessed quantitative differences in ATAC-seq signal intensities across the three stages. Primarily, a unified set of 91,746 peaks was generated (Supplementary Dataset 3) by merging accessible regions of individual stages and their genomic distribution is shown in Fig. 2b. The merged ATAC-seq dataset was used to identify differentially accessible chromatin regions (DACs), which illustrate the genome-wide temporal dynamics of chromatin accessibility in developing mouse forelimbs (Fig. 2c and Supplementary Fig. 2f). Pairwise comparisons between developmental stages viz. E9.75/E10.5 ($n = 30,694$), E10.5/E11.5 ($n = 22,178$), and E9.75/E11.5 ($n = 41,729$) identified potential DACs with consistent profiles among replicates (Fig. 2c). We found the largest differences in chromatin accessibility between E9.75 and E11.5, which parallels the observed transcriptome changes in forelimb buds (Fig. 1c, d and Supplementary Fig. 1e). These datasets allow an analysis of the temporal dynamics of chromatin accessibility and changes in transcript levels for key limb regulator genes such as *Gli3, Hand2*, and *Grem1* (Fig. 2d–f). Analogous to DEG modules (Fig. 1d, e), unbiased clustering using *k-means* resulted in six DAC modules with distinct temporal trajectories of chromatin accessibility (Fig. 2c, g). The robustness was validated by silhouette analysis (Supplementary Fig. 2g) and specific identifiers were assigned as defined for DEG modules. The majority of DACs belong to the E9.75$^{hi}$ module

(32.6%), followed by the E9.75$^{lo}$ (20%) and E11.5$^{hi}$ module (19.1%, Fig. 2h).

Next, the temporal correspondence between chromatin accessibility and gene expression was established by computing the pairwise likelihood of association between regions in each DAC module with genes of the DEG modules. This unbiased computational analysis reveals a striking temporal correspondence between DAC and DEG modules during mouse forelimb bud development (Fig. 2i and Supplementary Fig. 2h). Interestingly, we found that DAC modules contain a significantly higher number of accessible regions positionally associated with genes of the corresponding DEG modules. For instance, the DAC-E9.75$^{hi}$ module (Fig. 2g) contains a significantly higher number of accessible regions in proximity to genes belonging to the DEG-E9.75$^{hi}$ module (Fig. 1d). In addition, significant associations between DEG and DAC modules with similar rather than identical temporal trajectories are also apparent. For example, DAC-E9.75$^{hi}$ also shows a significant association with DEG-E11.5$^{lo}$ as the latter encompasses additional genes with the highest relative expression at E9.75 as discussed before (Figs. 2g, i, 1d). In contrast, no association between DAC and DEG modules with incoherent temporal trajectories is observed (Fig. 2i and Supplementary Fig. 2h). This analysis indicates that the temporal dynamics of transcription is synchronous with the accessibility of the positionally associated chromatin regions (indicated by black frames in Fig. 2i). The outcome suggests that the accessible chromatin regions participate in *cis*-regulation of the nearby genes and indicates that the developmental transitions during mouse forelimb development are reflected by the epigenome dynamics. These observations are in agreement with recent studies reporting that temporal regulation of chromatin accessibility is accompanied by changes in target genes activities in embryos of different species[8,30].

**Distinct temporal modes of gene expression and chromatin remodeling during chicken wing bud development.** Chicken wing buds are another paradigm to study the molecular mechanisms that govern vertebrate limb bud development. Therefore, we also investigated the chromatin remodeling and gene expression dynamics during chicken wing bud development to gain insight into chromatin accessibility mediated regulation. We performed ATAC-seq and RNA-seq profiling of chicken wing buds at Hamburger–Hamilton (HH) stages 20, 22, and late 24[32] (Fig. 3a). These stages are morphologically and molecularly equivalent to the orthologous mouse forelimb bud stages (E9.75, E10.5, and E11.5, Supplementary Fig. 3a, b). In particular, the conserved *Hoxa*, *Hoxd*, and SHH and BMP target genes display remarkably similar temporal gene expression dynamics during both mouse forelimb and chicken wing bud development (Supplementary Fig. 3c, d). In analogy to mouse forelimb buds, *k-means* clustering of the RNA- and ATAC-seq datasets (Supplementary Fig. 4a–d) identified six distinct DEG and DAC modules for developing chicken wing buds (Fig. 3b–d, Supplementary Fig. S4e–k, and Supplementary Dataset 4). The majority of DEGs belong to the HH20$^{lo}$ (31.5%) and HH20$^{hi}$ modules (25.8%, Supplementary Fig. S4j), while the majority of DACs are part of HH24$^{hi}$ (29.4%) and HH20$^{hi}$ modules (28.6%, Fig. 3e). We found the largest modulation of chromatin accessibility between HH20 and HH24 in chicken wing buds, which is in agreement with observed transcriptome and chromatin accessibility changes in mouse forelimb buds (Fig. 1d).

Next, the accessible chromatin regions were assigned putative target genes based on nearest proximity (Supplementary Dataset 4). In contrast to the mouse (Fig. 2i), the association analysis revealed no apparent temporal correspondence between chicken DEG and DAC modules (Fig. 3f). Instead, significant associations were observed between DAC and DEG

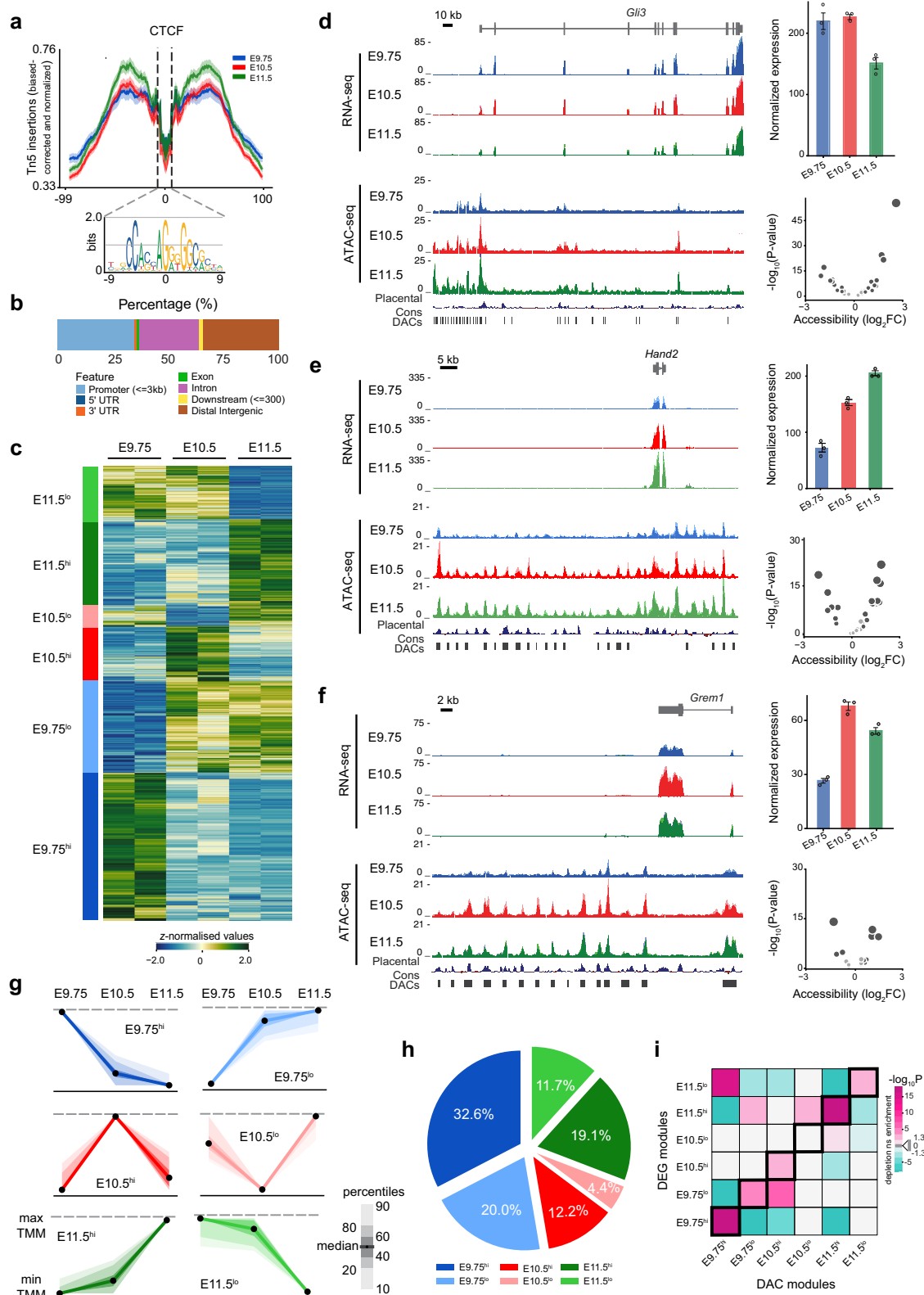

modules with similar temporal patterns. In particular, the DAC-HH20hi associates with DEG-HH22lo and DEG-HH24lo; DAC-HH20lo with DEG-HH24hi; DAC-HH24hi with DEG-HH20lo and DEG-HH22hi, and DAC-HH24lo with DEG-HH24lo and DEG-HH20hi (Fig. 3f and Supplementary Fig. 4l). These results indicate that the temporal profiles of chromatin accessibility and gene expression show correspondence during

the onset of wing bud (HH20) and autopod development (HH24), while their modulation is rather discordant during outgrowth (HH22).

**Comparative analysis of gene expression dynamics reveals both similarities and differences during mouse forelimb and chicken wing bud development.** Given the crucial role of differential

**Fig. 2 Temporal correspondence of DAC and DEG modules in mouse forelimb buds. a** CTCF footprint centered around CTCF binding sites for the different forelimb bud stages. Shown below is the CTCF consensus binding site. **b** Distribution of merged peaks in different genomic regions for mouse forelimb buds. **c** Heatmap shows the relative accessibility of DACs in mouse forelimb buds. DACs with similar temporal accessibility patterns across developmental stages were assigned to six different DAC modules (see panel **g**). **d**–**f** UCSC genome browser visualization (left panel) of the stage-specific RNA-seq and ATAC-seq profiles for the *Gli3* (**d**), *Hand2* (**e**), and *Grem1* (**f**) genomic landscapes. Significant DACs are indicated as vertical bars below the signal profiles. Right panels (top): bar plots showing normalized gene expression in the three forelimb bud stages. Error bars indicate the mean ± SEM among $n = 3$ independent biological replicates per developmental stage. Right panels (bottom): volcano plots showing statistical significance versus magnitude of fold change for accessible regions associated with the genes. **g** Scaled DAC module profiles were derived by *k-means* clustering of individual DACs. The DAC modules are indicated by the different color codes. Black dots represent the median values while the shaded areas signify the percentile ranges (indicated in the scale next to the graphs). **h** Pie chart depicting the distribution of DACs comprised within different DAC modules. **i** Heatmap showing the association of DAC and DEG modules. Significant associations were determined by calculating how frequently peaks in a particular DAC module were positionally associated with genes in each of the six DEG modules. Two-sided Fisher's exact test for enrichment was used to compute enrichment or depletion. The adjustments were made for multiple comparisons using Benjamini–Hochberg (BH) method and an adjusted *p* value < 0.05 was considered significant. The figure represents the log-transformed, adjusted *p* values from Fisher's exact test. Significant enrichments (odds ratio > 1) are shown in magenta and depletions (odds ratio < 1) in cyan. Light gray squares indicate non-significant associations (adjusted *p* value > 0.05). The corresponding DAC and DEG modules are outlined by black frames. DAC differentially accessible chromatin regions, DEG differentially expressed genes. Source data are provided as a Source Data file.

expression of conserved genes during evolutionary diversification of developing limbs[13], we investigated the extent to which gene expression diverges between mouse forelimb and chicken wing buds. To this end, the temporal expression trajectories of the mouse–chicken orthologues were compared to identify conserved and species-specific variations during forelimb and wing bud development (Fig. 4a). Of the 5406 DEGs identified by inter-stage comparisons of limb buds in at least one of the two species (Figs. 1c, d, and 3b, c), 3527 genes (65.24%) showed a one-to-one orthologous correspondence between the mouse and chicken. These 3527 genes define an orthologous set of DEGs (OSD) shared between mouse and chicken limb buds that display significant temporal variation in either or both species (Fig. 4a). Of all genes in the OSD, 7% are associated with mouse limb congenital malformations (Supplementary Dataset 5). Moreover, the OSD includes genes that are differentially expressed either specifically in mouse (63%, mm-only) or chicken limb buds (17%, gg-only, Supplementary Dataset 5). The others are DEGs in both species (20%, mm-gg, Supplementary Dataset 5) and the majority of them display divergent gene expression trajectories (Supplementary Fig. 5a). Concurrently, DEGs with similar expression kinetics in both species might function as part of evolutionary conserved gene regulatory networks (GRNs) that control vertebrate limb bud development.

To gain insight into the underlying transcriptional control mechanisms, we identified developmentally relevant TFs (11%) within the OSD. The most represented TF families include homeobox (mm-gg: 28.3%, mm-only: 24%, gg-only: 21.3%), zinc finger (zf-C2H2, mm-gg: 15.0%; mm-only: 13%, gg-only: 8.5%) and bHLH family members (mm-gg: 12.6%, mm-only: 10%, gg-only: 12.8%; Fig. 4b and Supplementary Dataset 5). Previous studies have reported essential roles for homeobox genes (e.g., *Hox*, *Meis*, and *Dlx* genes) during limb bud patterning and outgrowth[28,33,34]. In addition, zf-C2H2 zinc finger proteins (*Sp* and *Klf* gene family members) are required for establishing AER signaling[35], while bHLH transcriptional regulators are essential for limb bud mesenchymal patterning and proliferation[15]. In contrast to mm-only DEGs, forkhead TFs represent the third largest family among gg-only DEGs (10.6%, Fig. 4b), which function in the specification of mesenchymal progenitors during limb bud development[36]. Other functionally relevant TFs families include *Cut*, *Ap2*, *Runt*, *Hmg-box*, *Pax*, and *T-box* genes (Fig. 4b).

Furthermore, genes active in limb bud mesenchymal progenitors (BMP pathway: *Smoc1* and *Msx1*) and osteo-chondrogenic progenitors (*Twist2*, *Sox9*, and *Runx2*) share similar temporal expression profiles in both species, which points to the

conservation of the underlying morpho-regulatory processes (Fig. 4c). In contrast, *Adamts9* and *Acta2*, which mark connective tissue and smooth muscle progenitors have complementary temporal expression profiles in mouse and chicken limb buds (Fig. 4c). Finally, TFs of the myogenic lineage such as *Myod1*, *Myf5*, *Pax7*, and *Eya2* display high expression from stage HH22 onward in chicken wing buds, while most of them are upregulated at E11.5 in mouse forelimb buds (Fig. 4c). Together, the similar and the species-specific expression trajectories of key regulators reveal the existence of conserved and heterochronic developmental processes in mouse and chicken limb buds (Fig. 4c and Supplementary Fig. 5b, c).

To determine the extent by which quantitative temporal changes could be paralleled by spatial expression changes in limb buds, we analysed the spatial expression of a few key regulator genes in both species by whole-mount RNA in situ hybridization (Supplementary Table 1). Overall, the spatial distributions are similar during early limb development in both species (E9.75/HH20), while for a few genes species-specific differences are apparent in the developing autopod primordia (E11.5/late-HH24, Supplementary Fig. 6). In particular, the expression of the *Hand2*, *Tbx2*, and *Hoxd13* transcriptional regulators appears more posteriorly restricted in the developing chicken (late-HH24) than mouse autopod (E11.5, arrowheads, Supplementary Fig. 6a–c). Furthermore, the expression of the BMP target *Msx2*[19] is increased in the anterior sub-ectodermal mesenchyme in chicken wing bud compared to mouse forelimb buds (arrowheads, Supplementary Fig. 6d). In contrast, no striking spatiotemporal changes were observed for *Pkdcc* kinase[37] and the TF *Prdm1* (Supplementary Fig. 6e, f). This analysis shows that variations in transcript levels can be paralleled by species-specific spatial changes in gene expression during mouse forelimb and chicken wing development.

**Dynamic TF–DNA interaction within accessible chromatin regions during forelimb and wing bud development.** Our transcriptome analysis revealed significant differences in gene expression during mouse forelimb and chicken wing bud development (Figs. 1, 3), which could be a likely consequence of the dynamic interactions of TFs with their binding regions in accessible chromatin[6,38]. To this end, we used a curated list of experimentally derived TFBS motifs[39] to identify enriched TFBS within DAC modules of both species (Fig. 5 and Supplementary Fig. 7). In particular, a comparison of hi DAC modules with dissimilar accessibility patterns (Supplementary Fig. 7a, b) revealed significantly enriched TFBS in both species (Fig. 5a). Interestingly, DACs with high accessibility in early limb buds

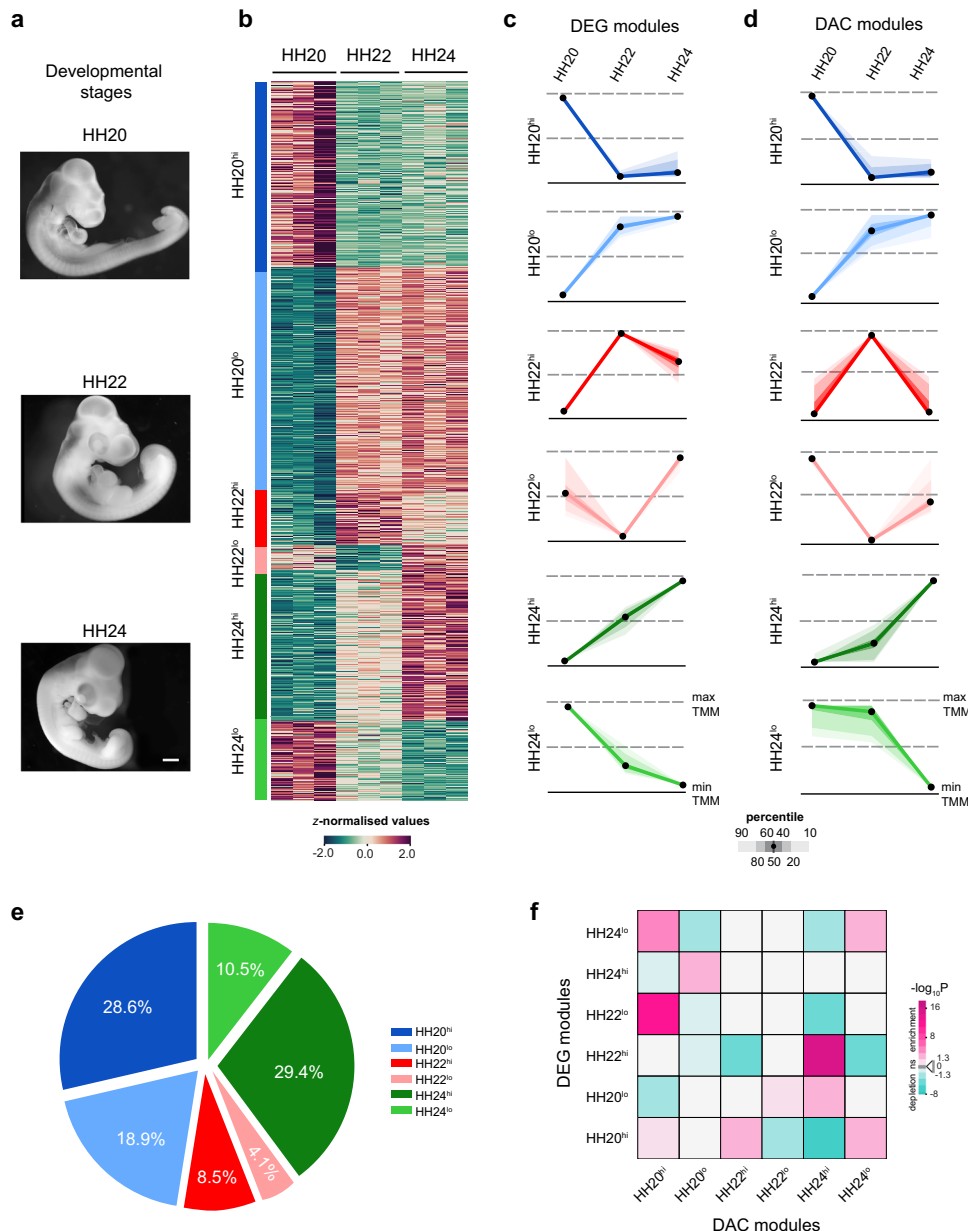

**Fig. 3 Temporal dynamics of chromatin accessibility and gene expression in chicken wing buds. a** Illustrations of chicken embryos at the developmental stages (HH20, HH22, and late-HH24) that were used for the analysis of wing bud development (scale bar: 250 μm). **b** Heatmap illustrates relative gene expression trajectories across the developmental stages for all DEGs. **c**, **d** Temporal profiles of the six DEG modules (**c**) and DAC modules (**d**). Scaled module profiles were derived by *k-means* clustering. The color code is the same for DEG and DAC modules. The black dots represent the median values while the shaded areas signify the percentiles ranges indicated (scale below the graphs). **e** Pie chart depicting the fraction of DACs assigned to each of the six DAC modules. **f** Heatmap showing the association of DAC with DEG modules. Significant associations were determined by calculating how frequently peaks in a DAC module are positionally associated with genes of the six different DEG modules. Two-sided Fisher's exact test for enrichment was used to compute enrichment or depletion. The adjustments were made for multiple comparisons using Benjamini–Hochberg (BH) method and an adjusted *p* value < 0.05 was considered as significant. The figure represents the log-transformed, adjusted *p* values from Fisher's exact test with significant enrichments (odds ratio > 1) shown in magenta and depletions in cyan (odds ratio < 1). Light gray squares show non-significant associations (adjusted *p* value > 0.05). DAC differentially accessible chromatin regions, DEG differentially expressed genes. Source data are provided as a Source Data file.

(E9.5$^{hi}$, HH20$^{hi}$) are enriched for binding motifs of zinc finger (GLI, SP, ZEB, and EGR), bHLH (HAND2, MYCN, HES1, and HEY2), GATA, and T-box TF families. This is in agreement with their functions during the onset of limb bud development, axis polarization, and early outgrowth (Fig. 5a and Supplementary Fig. 7c)[15]. The TFBS of homeobox family member MEIS2 is significantly enriched at early stages (E9.75/HH22, Fig. 5a), while there is no preferential enrichment of HAND2 binding motifs in chicken wing buds. During subsequent limb bud outgrowth and autopod formation, the binding sites for transcriptional regulators such as TWIST2, homeodomain-encoding TFs (LMX, MSX, LHX, DLX, and HOX13), and HMG-box members such as SOX9 are enriched in both species (Fig. 5a and Supplementary Fig. 7c). Furthermore, the DAC$^{hi}$ and DAC$^{lo}$ modules of corresponding stages display rather complementary patterns of TFBS enrichment (Supplementary Fig. 7c).

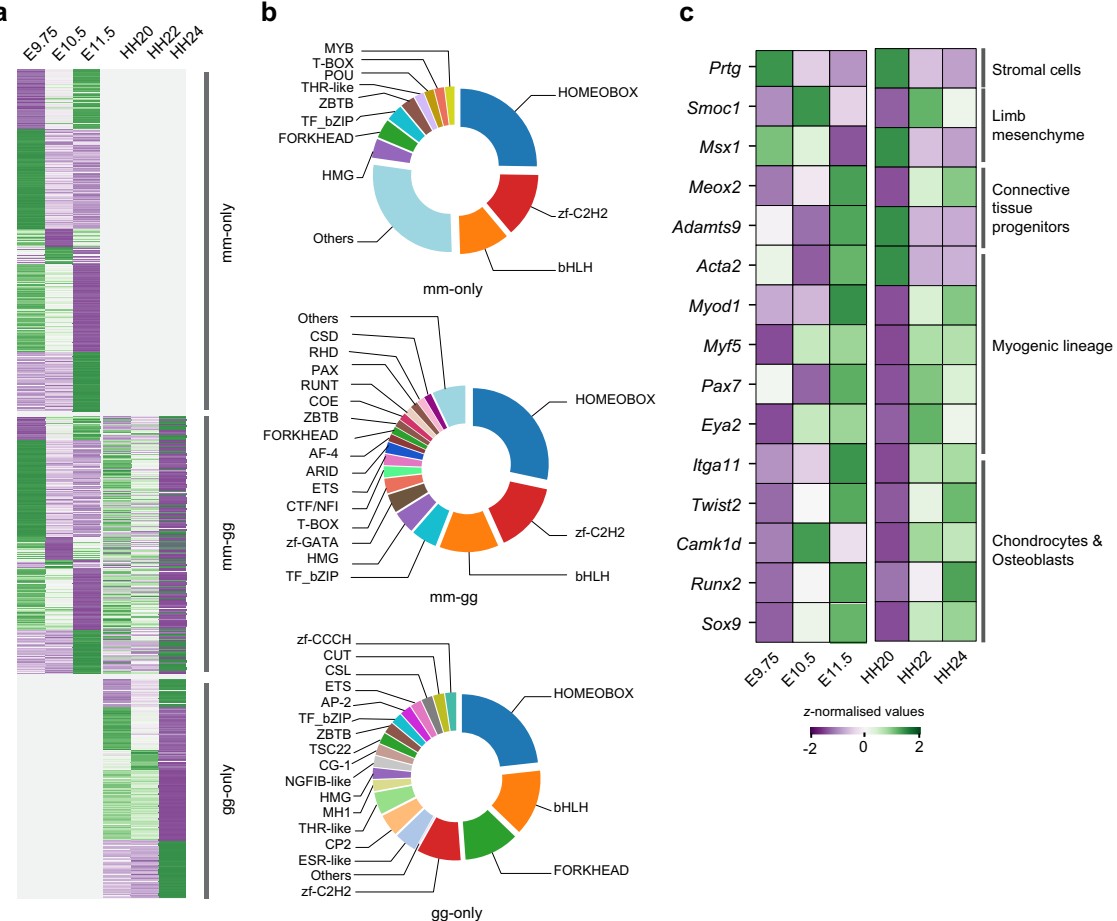

**Fig. 4 Comparative analysis of the gene expression dynamics in limb buds reveals shared and species-specific variations. a** Heatmap showing the relative expression of genes in the orthologous set of DEGs (OSD, n = 3527). The genes displaying significant temporal changes in mouse forelimbs (mm-only), chicken wing buds (gg-only), and limb buds of both species (mm-gg). **b** Distribution of key TF family members in different categories of OSD. Different TF families are indicated by unique colors. TF families with ≤1.5% were pooled into the "Others" category. **c** Heatmap showing *z-normalized* expression of different key markers during mouse forelimb and chick wing bud development that function in specific developmental processes (indicated on the right). Source data are provided as a Source Data file.

To delve further into the accessibility of motifs at single base-pair resolution, we inferred computational TF footprints using species-specific ATAC-seq profiles (Fig. 5b). In agreement with binding site enrichments, the deeper footprints of GLI3 and HAND2 (upper panels in Fig. 5b and Supplementary Fig. 7d) corroborate their preferential binding activity during the onset of mouse forelimb bud development (E9.75). In chicken wing buds, the deepest GLI3 footprint is observed at the earliest stage analysed (HH20), while HAND2 footprints are similar across stages (lower panels in Fig. 5b). The MEIS transcriptional regulators are essential for establishing proximal limb bud identity and their spatial distribution forms a temporally dynamic PD gradient during outgrowth[34,40]. The overall shallow MEIS2 footprints in mouse and chicken limb buds indicate that MEIS2 interacts rather transiently with its binding sites in open chromatin (Fig. 5b and Supplementary Fig. 7d), which could be a direct consequence of its dynamic distribution. By contrast, sharp footprints for HOXD13, an essential regulator of limb and digit development[28] are detected in both species (Fig. 5b and Supplementary Fig. 7d). Together, this analysis shows that the dynamic patterns of TFBS enrichment and computational footprints of select key TFs are remarkably well conserved despite the fact that *Mammalia* (mouse) and *Sauropsida* (chicken) diverged ~330 million years ago. However, divergent species-specific TFBS enrichment and footprint profiles were

detected for key regulators such as GATA6, MECOM, and the WNT signaling mediators LEF1 and TCF7 (Fig. 5a, c and Supplementary Fig. 7c, d).

**In silico identification of candidate target genes of HAND2 and GLI3 in mouse forelimb and chicken wing buds**. The initial AP polarization occurs during the onset of mouse limb bud development under the control of the GLI3R and HAND2 transcriptional regulators (Fig. 6a) and the GRNs controlling the establishment of AP identities are conserved[15]. Therefore, we identified the regulatory interactions mediated by GLI3 and HAND2 in limb buds using a limited set of genes with essential functions during patterning and outgrowth (n = 160 genes, Supplementary Dataset 6). Integration of chromatin accessibility, TFBS information, and gene expression in both species reveal a range of shared and species-specific candidate target genes for GLI3 and HAND2 (Fig. 6b and Supplementary Fig. 8). The predicted target genes include HAND2 and GLI3 targets that have been verified by previous genetic and molecular analysis in mouse limb buds[16–18,41–44] (indicated by asterisks in Fig. 6b). These results underscore the validity of the chosen approach[38] for identifying candidate target genes in both species (Fig. 6b and Supplementary Fig. 8). Moreover, this analysis further expands the number of potential targets of GLI3 and HAND2 by

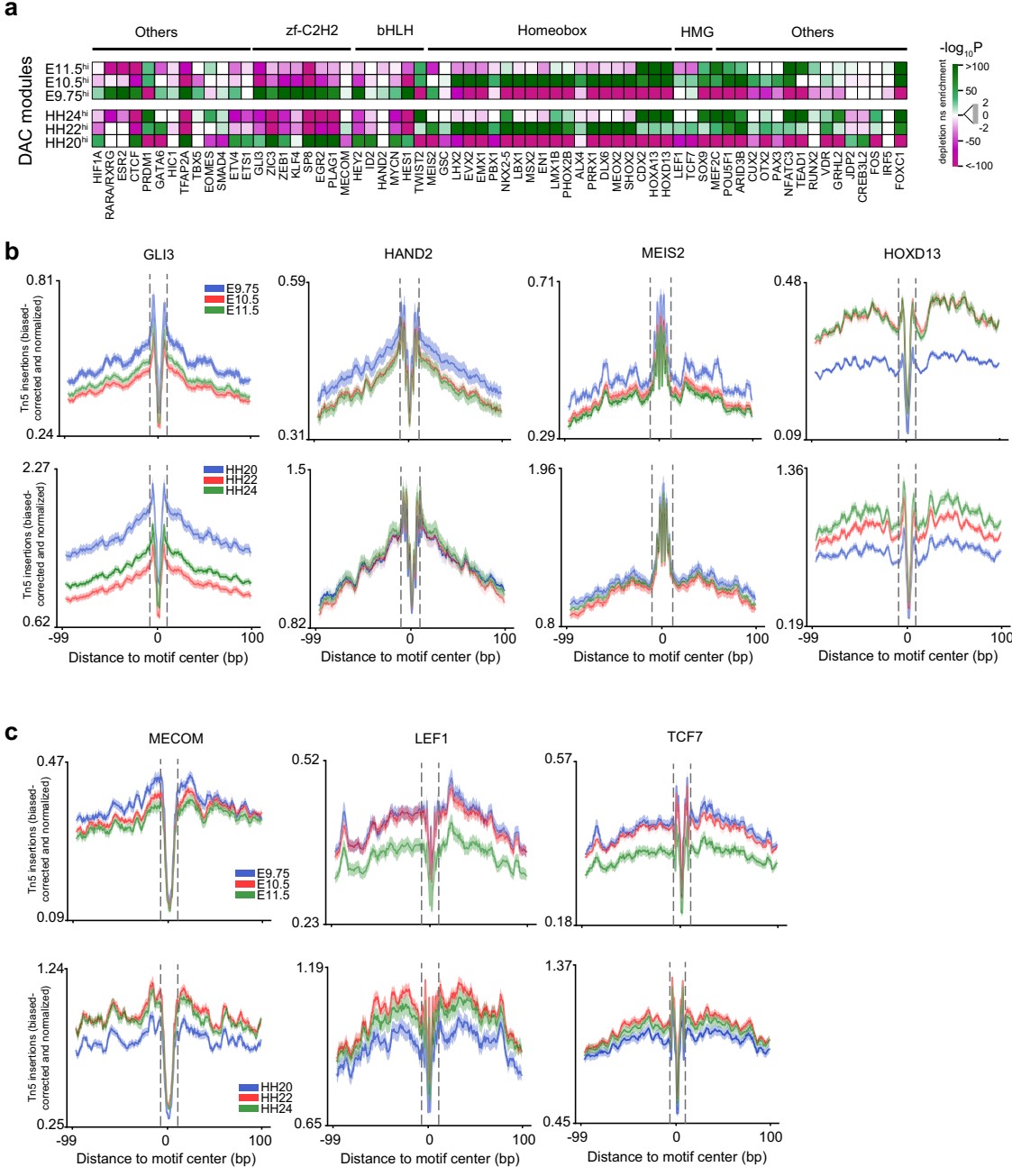

**Fig. 5 In silico prediction of TF binding activity within DACs reveal stage-specific TF–DNA interaction dynamics in both species. a** Heatmap showing significantly enriched or depleted TFBS within DAC-hi modules of mouse forelimb (top panel) and chicken wing buds (bottom panel). For each DAC module, significant enrichment (green) and depletion (magenta) of TF motifs were calculated against unrelated modules with complementary accessibility patterns using analysis of motif enrichment (AME, Supplementary Fig. 7a, b, Methods). **b** Visualization of the ATAC-seq footprints for key transcription factors that function in AP (GLI3 and HAND2) and PD (MEIS2 and HOXD13) axes patterning during mouse (upper panel) and chicken limb bud development (lower panel). The average ATAC-seq signal around footprints is shown for select TFs at the three limb bud stages (indicated by different colors). **c** Shown are the dynamic patterns of the MECOM, LEF1, and TCF7 footprints in mouse and chicken limb buds. Dashed lines demarcate the consensus TF binding motif regions and their sequence LOGOS showing information content at each nucleotide are shown in Supplementary Fig. 7d. DAC differentially accessible chromatin regions, TFBS transcription factor binding sites. Source data are provided as a Source Data file.

identifying additional interactions with key TFs that function in limb bud patterning (Fig. 6b). In fact, the identification of HAND2 and GLI3 candidate target genes in both species is an asset given the lack of corresponding ChIP-seq datasets in chickens.

Intriguingly, the majority of candidate target genes for the two TFs are shared, which points to their transcriptional co-regulation by GLI3 and HAND2 in mouse forelimb buds (gray lines,

Fig. 6b). Moreover, GLI3- and HAND2-specific interactions were also detected (GLI3: ~22%; HAND2: ~17%; magenta lines in Fig. 6b). Similarly, the candidate HAND2 and GLI3 target genes identified in chicken wing buds include both co-regulated (gray lines, Supplementary Fig. 8) and specific candidate target genes (GLI3: ~31%; HAND2: ~49%, magenta lines, Supplementary Fig. 8). Furthermore, 35% of the GLI3 and 50% of the HAND2 target genes are conserved in both species, while the others are

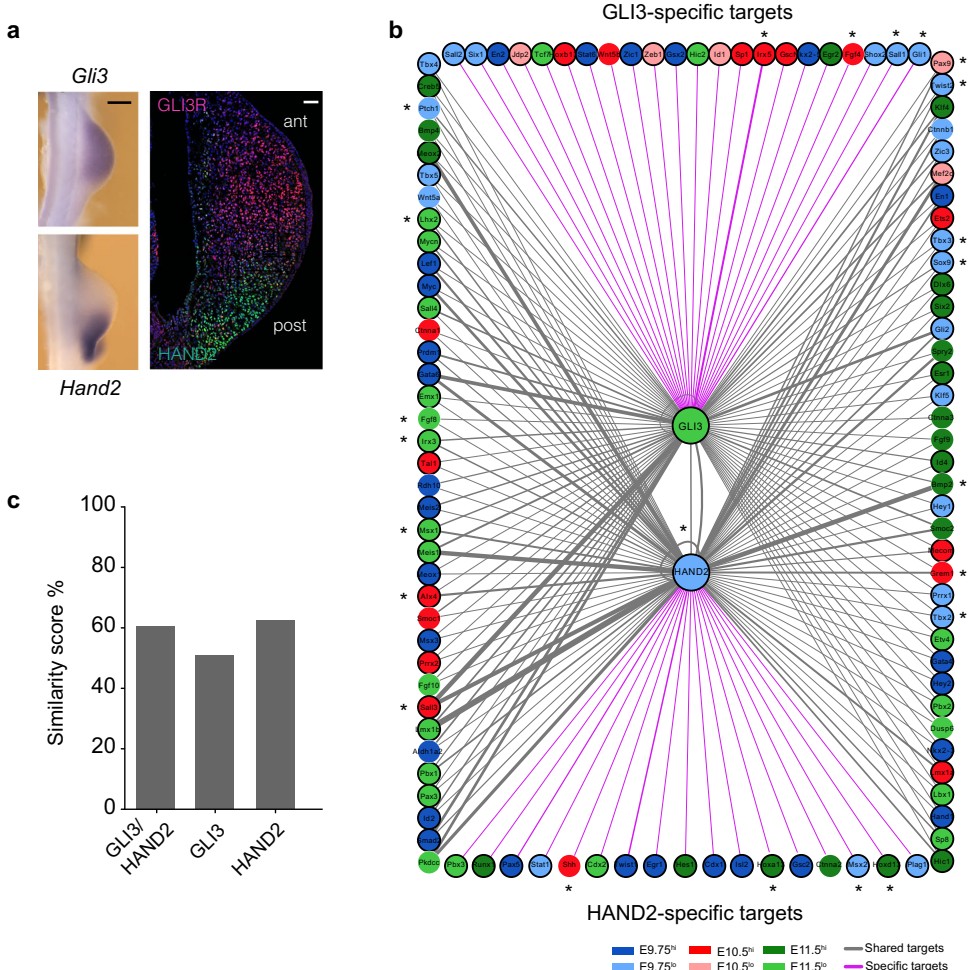

**Fig. 6 In silico identification of candidate GLI3 and HAND2 target genes in mouse forelimb buds. a** Left Panel: whole-mount RNA in situ hybridization showing the spatial distribution of *Gli3* ($n = 3$, upper-left panel) and *Hand2* transcripts ($n = 3$, lower-left panel) in mouse forelimb buds at E9.75 (scale bar: 250 μm). Right panel: colocalization of HAND2 (green) and GLI3R (red) proteins in mouse forelimb buds (histological section at E9.75, reproduced from ref. [17] with permission from *Developmental Cell*). Scale bar: 50 μm. **b** Computational identification of *cis*-regulatory interactions of GLI3 and HAND2 with candidate target genes in limb buds. Circles represent nodes while the interactions are represented by solid lines (edges). GLI3 and HAND2 are source nodes (shown in the center) and their candidate target genes are genes linked to accessible regions with GLI3 and/or HAND2 TF binding motifs. The edges of the network represent motifs and are weighted (thickness of the line) based on the number of TF binding motifs within the accessible regions linked to the target gene, which support a connection. GLI3 and HAND2-specific interactions are shown by magenta-colored lines while shared interactions are represented by gray lines. The color code of the candidate target genes corresponds to their closest DEG module based on the correlation of their gene expression with cluster centroids. Target genes encoding transcriptional regulators are outlined in bold. Molecularly and/or genetically verified GLI3 and HAND2 target genes are indicated by asterisks. **c** Bar plot showing the similarity score as a percentage of the GLI3 and HAND2 mediated interactions shared between developing mouse forelimb and chicken wing buds (see also Supplementary Fig. 8). DEG differentially expressed genes. Source data are provided as a Source Data file.

unique to either species. Considering both source TFs and their interacting partners, a comparison of the GLI3-HAND2 interactions yielded an overall network similarity score of 60.6% between both species (Fig. 6c). These common targets and their interactions could be part of evolutionary conserved GRNs that function in AP axis polarization and the onset of outgrowth. By contrast, the non-conserved interactions might be part of diversified regulatory mechanisms underlying the development of species-specific traits.

**CARs exhibit divergent enhancer activity indicative of molecular differences in gene regulation.** Recent studies have shown that morphological diversification of vertebrate limb bud development is paralleled by functional genomic alterations of the CREs encoding transcriptional enhancers[11,12]. To identify highly diversified *cis*-regulatory regions in the chicken genome, we first

assembled a comprehensive set of mouse limb bud enhancers previously discovered using *lacZ*[45], CAGE[46], and Capture-C[47] that are part of accessible chromatin (Supplementary Fig. 9a). Using a comparative genomics approach[12], 314 CARs with significantly accelerated sequence substitutions (FDR < 5%) were identified (Supplementary Dataset 7). These CARs are orthologous to active mouse limb bud enhancers and highly conserved across different vertebrates. A previous study identified so-called avian-specific highly conserved elements (ASHCEs) by comparative analysis of 48 avian species[48]. These ASHCEs are highly conserved among avian species, but not in other vertebrate clades. The majority of CARs (96%) identified in our study are not ASHCEs, suggesting that CARs are indeed chicken-specific (Supplementary Dataset 8). By contrast, most CARs (78%) and DACs (90%) overlap with putative chicken embryonic and/or limb enhancers previously predicted using combinatorial patterns

of epigenetic signatures[48], which corroborates their potential roles in *cis*-regulation (Supplementary Fig. 9b, Supplementary Dataset 8, 9, and Supplementary Tables 2, 3). Among the different predicted enhancer activity states, strong enhancers are over-represented among CARs (48.40%, Supplementary Fig. 9b and Supplementary Table 2). We also mapped the extent to which these CARs overlap with differently accessible chromatin in limb buds (Supplementary Fig. 9c–e and Supplementary Dataset 10). Of all CARs, 39.5% are differentially accessible in chicken wing buds (CAR-gg-DAC and CAR-mm/gg-DAC) while the others display differential accessibility only for the mouse orthologues (21%, CAR-mm-DAC) or in neither of the two species (39.5%, CAR-no-DAC, Supplementary Fig. 9c–e). Assigning putative target genes to the CARs by proximity mapping showed that 146 CARs (~47%) reside within 1 Mb of a developmental regulator gene (Supplementary Dataset 7). Furthermore, about half of all CARs are associated with at least one putative target gene that displays significant temporal differences in expression, i.e., is a DEG in either or both species (Supplementary Fig. 9d–h and Supplementary Dataset 10).

In analogy to the bat accelerated regions[12], we hypothesized that CARs might contribute to chicken-specific *cis*-regulation in limb buds. Therefore, the enhancer activity of select CARs and their mouse orthologues linked to genes functioning during limb bud development were assessed by *lacZ* reporter assays in transgenic mouse embryos (Fig. 7 and Supplementary Fig. 10b, c). In addition, we examined the spatial expression of the associated target gene in limb buds of both species. This comparative analysis revealed striking differences in enhancer activities between CARs and their mouse cognate enhancers while changes in the spatial expression of the associated target genes were at most subtle (Fig. 7, Supplementary Fig. 10 and Supplementary Table 4). For example, CAR134 and its cognate mouse enhancer are linked to *Tbx2* and display divergent *lacZ* activities (panel *lacZ*, Fig. 7a). The activity of the cognate mouse enhancer is apparent in anterior and posterior domains, while CAR134 activity is restricted to the posterior limb bud mesenchyme. These differential enhancer activities reflect the endogenous *Tbx2* expression, as CAR134 overlaps the posterior *Tbx2* domain in chicken wing buds, while the mouse enhancer overlaps the distal-anteriorly expanded posterior and the anterior Tbx2 domain in the mouse forelimb buds (arrowheads, panel *Tbx2*, Fig. 7a). In addition, both enhancers are active in the *Tbx2*-expressing branchial arches (white arrowheads, panel *lacZ*, Fig. 7a)[49]. CAR117 and its mouse orthologue are linked to the inhibitory BMP/TGFß pathway regulator *Smad7* and *lacZ* reveals their divergent activities in transgenic mouse limb buds (panel *lacZ*, Fig. 7b). The mouse cognate enhancer is active in the anterior mesenchyme, while CAR117 is active in the anterior forelimb and broader anterior and posterior domains in hindlimb buds. *Smad7* expression in mouse limb buds is low and diffuse, while in chicken wing and legs buds, *Smad7* expression is detected broadly in the sub-ectodermal mesenchyme encompassing distinct anterior and posterior domains (arrowheads, panel *Smad7*, Fig. 7b)[50]. In particular, the CAR117 activity in mouse hindlimb buds recapitulates aspects of the anteriorly and posteriorly enhanced *Smad7* expression in chicken limb buds. CAR97 and its cognate mouse enhancer are located close to the *Prrx2* gene and broadly active, with mouse enhancer activity being more distally restricted within the posterior limb bud mesenchyme (panel *lacZ*, Fig. 7c). In both species, *Prrx2* is expressed throughout the undifferentiated peripheral limb bud mesenchyme without apparent differences (panel *Prrx2*, Fig. 7c). Other CARs display weak or no reproducible enhancer activity compared to their mouse orthologues. In particular, CAR22 and its mouse orthologue are located in the *Gli3* genomic

landscape (Fig. 7d). The robust activity of the mouse enhancer overlaps the endogenous *Gli3* expression, while CAR22 activity is low and variable (panel *lacZ*, Fig. 7d). No reproducible *lacZ* activity is detected for the CARs linked to the BMP target genes *Id1* (CAR50) and *Msx1* (CAR74), which contrasts the robust activity of the cognate mouse enhancer (Supplementary Fig. 10b, c). Despite these significant differences in enhancer activities, no major spatial changes in *Gli3, Id1*, and *Msx1* expression are apparent in limb buds of both species at E11.5/HH24 (Fig. 7d and Supplementary Fig. 10b, c).

## Discussion

Developmental stage-specific transcriptome profiling identified DEG modules characterized by distinct expression trajectories, which reveals the temporal gene expression dynamics during limb bud development. Concurrently, we observed widespread chromatin remodeling resulting in dynamic opening and closing of chromatin regions. In mouse limb buds, the coupling of DACs and DEGs showed significant stage-specific associations, which reveals the remarkable synchrony of chromatin remodeling with temporal changes in gene expression. By contrast, the chicken DAC modules showed association with those DEG modules with similar rather than identical temporal kinetics, which points to apparent heterochrony during chicken wing bud outgrowth. A recent genome-wide analysis of medaka embryos also uncovered nonsynchronous modes of chromatin remodeling and gene expression[29]. In medaka embryos, opposing dynamics of chromatin accessibility and corresponding genes point to repressive modes, while enhancer switching and opening of chromatin prior to gene activation result in asynchrony and heterochrony between chromatin remodeling and transcriptional regulation[29]. Similarly, the opening of chromatin precedes activation of the associated genes during the onset of gene expression in vertebrate embryos[7,51,52]. It appears that the gene expression dynamics in mouse limb buds are likely regulated via changes in nearby chromatin accessibility. Given that, ~90% of peak-gene assignments based on proximity are likely correct[53], our association analysis points to functional interactions between the putative regulatory regions and associated genes in mouse limb buds. By contrast, the observed heterochrony between chromatin accessibility and gene expression in chicken wing buds could possibly be explained by the higher impact of repression and/or enhancer switching compared to mouse limb buds[29]. Our analysis indicates that spatiotemporal gene expression is in general more conserved between both species during onset (E9.75/HH20) rather than the progression of the limb bud and autopod development, which could be due to molecular diversification during outgrowth. Such spatiotemporal divergence during limb bud outgrowth is reminiscent of the molecular analysis of expression changes in artiodactyl limb buds, which have been proposed to underlie the evolutionary diversification from pentadactyl limbs and the appearance of artiodactyl limb traits[9,10,54].

Changes in chromatin accessibility impact gene regulation by controlling the *cis*-regulatory interactions of TF complexes with their DNA binding sites, which ultimately leads to chromatin remodeling[1,55,56]. Our unbiased enrichment analysis of TF motifs within DAC regions reveal TF–DNA interactions such that distinct TF binding motifs gained or lost accessibility at specific developmental stages. In agreement with their functions, the binding sites for TFs active during the onset of limb bud development are preferentially enriched in accessible chromatin regions at early stages. By contrast, binding sites of TFs functioning during outgrowth and autopod formation are enriched later in accessible chromatin regions. Furthermore, the computational footprinting of TFs corroborates this TFBS enrichment

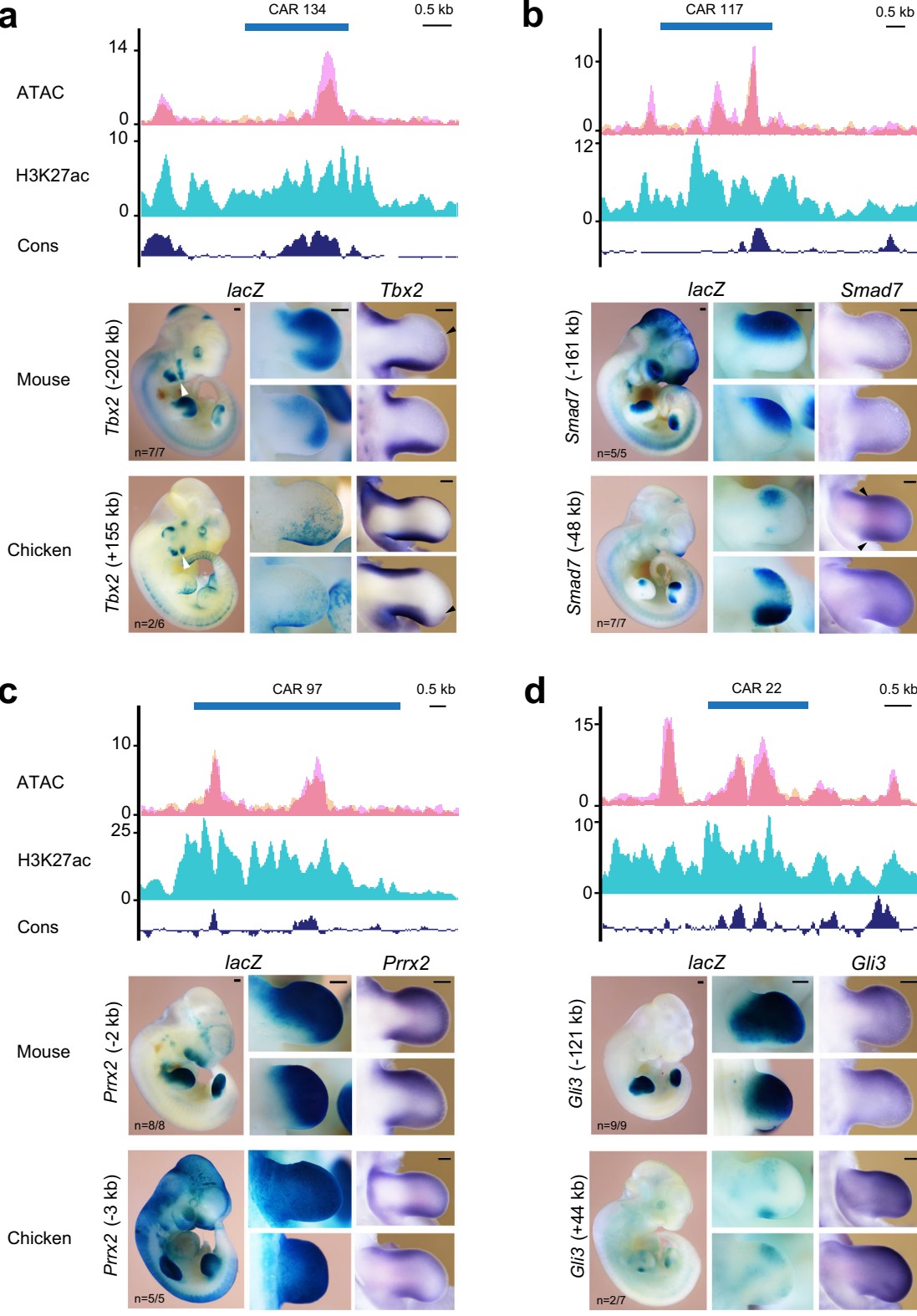

analysis. Together, these observations suggest that chromatin remodeling exerts regulatory capacity by facilitating developmental time-restricted TF–DNA interactions, which may impact the transcriptional regulation of the associated genes. A recent analysis of bovine preimplantation embryos shows that the opening of chromatin regions enriched in maternal TFBS precedes activation of embryonic transcription[7]. In medaka embryos, the temporal profiles of chromatin accessibility and predicted interactions with pioneer TFs match the dynamics of gene expression[29]. Furthermore, our analysis of the TF–DNA interactions in mouse and chicken limb buds reinforces the notion that gene regulatory properties including TF binding and chromatin states are remarkably well conserved among evolutionarily distant species irrespective of DNA sequence conservation[57]. This observation also points to possible canalization of the underlying epigenetic and *cis*-regulatory mechanisms across species[58].

**Fig. 7 Mouse *lacZ* transgenic assays reveal the divergent enhancer activity of CARs and their mouse orthologues. a–d** ATAC-seq (magenta) and histone H3K27ac (cyan) signals reveal that the mouse orthologues of CARs are part of active chromatin in mouse forelimb buds at E10.5. ATAC-seq signals from replicates are represented by two distinct shades namely peach (RGB: 246, 202, 160) and magenta (RGB: 246, 160, 243), while the regions overlaying accessibility signals from both replicates appear as dark pink (RGB: 235, 127, 153). Blue bars indicate the mouse regions orthologous to the CARs. *lacZ* panels: the enhancer activity of CARs and their mouse orthologous assessed by transgenic mouse *lacZ* reporter assays. The distance between the candidate enhancer region and the transcription start site of the associated gene is indicated. CAR134 and its mouse orthologue are associated to the *Tbx2* gene (panel **a**), CAR117 and its mouse orthologue are linked to *Smad7* (panel **b**), CAR97 and its mouse orthologue are linked to *Prrx2* (panel **c**), and CAR22 and its mouse orthologue to *Gli3* (panel **d**). For all panels, a representative transgenic mouse embryo (E11.0–E11.5, *lacZ* panel) and enlargements of its fore- and hindlimb buds are shown (*lacZ* panel). $n =$ fraction of transgenic embryos with limb bud expression. In situ panels: spatial expression of the associated gene in mouse and chicken fore/wing and hindlimb buds at orthologous stages (E11.5/HH24). Black arrowheads indicate the anterior *Tbx2* expression boundaries in mouse forelimb and chicken wing buds (panel **a**) and the distinct anterior and posterior *Smad7* domains in chicken wing buds (panel **b**). White arrowheads point to *Tbx2* expression in branchial arches. CAR chicken accelerated region, scale bars: 250 μm.

Our integrative analysis of chromatin accessibility, gene expression, and TF motif enrichment provides fundamental insights into the key interactions mediated by GLI3 and HAND2 in mouse and chicken limb buds. Unlike mouse, few TF-target genes have been identified for chicken or non-model species due to the lack of specific antibodies or epitope-tagged alleles for TFs of interest. A notable exception is the identification of SOX9 and HOXA13 target genes during chicken autopod development and chondrogenesis[59,60]. Therefore, our in silico analysis serves as a powerful complementary approach[38] for rapid identification of the possible range of target genes for TFs of interest in (non-model) species. Our proof-of-principle analysis of TF-target predictions yielded both co-regulated and specific targets of GLI3 and HAND2 that include the previously identified and experimentally verified key targets in mouse limb buds[16–18,41–44]. These findings remarkably expand the GLI3 and HAND2 network by incorporating additional interactions, especially for chicken limb buds where no direct targets have been reported. In addition to species–species interactions, the majority of the predicted target genes are shared between limb buds of both species. This corroborates the evolutionary conservation of the HAND2-GLI3 target gene network that controls axis polarization during the onset of tetrapod limb bud development[9,10]. In chicken limb buds, targeted overexpression and/or genome engineering approaches[61,62] could be used to dissect the functional consequences of altering the predicted TF-target gene interactions. Our analysis is proof-of-principle that these datasets are validated resources for future comparative analysis of chicken and mouse limb bud development and computational identification of TF-target gene networks as has been proposed for other biological systems[6,38,63].

Finally, we identified CARs whose sequences have significantly diverged during chicken evolution in comparison to other vertebrates. Transgenic analysis of selected CARs reveals the divergence of limb bud activities from their mouse orthologues. Recent comparative analysis of the *HoxD cis*-regulatory landscape in mouse and chicken limb buds showed that the sequence and activity of the chicken CS93 enhancer differ from its mouse orthologue[24], but shares similarities with a bat enhancer that was previously identified as bat accelerated region BAR116[12]. The limb bud activities of these orthologous enhancers in mouse, chicken, and bat correlate with the observed differences in endogenous *HoxD* gene expression in these species[24]. Conversely, no spatial changes in gene expression are apparent in other cases, which is not unexpected as genetic analysis established that enhancer redundancy provides developmental gene expression with *cis*-regulatory robustness[64]. Most CARs analysed in this study display divergent activities in transgenic fore- and hindlimb buds in comparison to their mouse cognate enhancers, which often is not paralleled by spatial expression changes of the associated target gene. Interestingly, for CAR117 the chicken-specific

changes in enhancer activity correlate with the spatial differences in *Smad7* expression during chicken limb bud development (this study). This may be of functional relevance as the inhibitory *Smad7* is a target of BMP/TGFß signaling transduction[50] and the BMP pathway functions in shaping the developing limb skeleton including digits[14]. Overall, our comprehensive comparative analysis between mouse forelimb and chicken wing bud development provides molecular insights into shared and species-specific regulation which can be an asset for future Evo-Devo studies.

## Methods

**Ethics statement and approval of all animal experimentation**. All animal experiments were performed in accordance with Swiss laws and approved by the regulatory and ethic committees/authorities of the Regional Commission on Animal Experimentation and the Cantonal Veterinary Office of the city of Basel under a mandate from the confederation (mouse and chicken embryos). The 3R principles were implemented in all animal study designs and power calculations were performed and/or set standards for respective experimental analysis implemented to ensure reproducibility. The *lacZ* reporter analysis was performed blinded as embryos were stained and results scored prior to genotyping of the founder embryos. Embryos of both sexes were included in all analyses.

**Mice and embryo collection**. Mice were housed in individually ventilated cages (Greenline-Tecniplast) at 22 °C, 55% humidity, and a light cycle of 12:12 with 30 min sunrise and sunset. Mouse embryos were collected from timed matings at the appropriate embryonic days E9.75 (28–30 somites), E10.5 (34–36 somites), and E11.5 (43–45 somites). For chicken, fertilized White Leghorn eggs (Animalco, Switzerland) were incubated at 38 °C, 55% humidity, and embryos were isolated at the Hamburger–Hamilton stages HH20, HH22, and HH24[32]. Forelimb and wing buds were used for RNA-seq, ATAC-seq, and embryos for comparative whole-mount in situ analysis.

**ATAC-seq library construction**. ATAC-seq experiments were performed according to standard protocols[31]. Briefly, pairs of forelimb and wing buds at the stages analysed were dissected in ice-cold 1x phosphate-buffered saline (PBS). Samples were collected as biological replicates and limb bud cells were isolated using the Dounce Tissue Grinder Pestles A and B for 20 strokes each in 1 ml cold PBS. Cells were counted using a Neubauer counting chamber and $75 \times 10^5$ cells per sample were homogenized in lysis buffer (10 mM Tris-HCl pH 7.4, 10 mM NaCl, 3 mM MgCl₂, 0.1% (v/v) Igepal CA-630). Thus, potential bias due to the significant increase in cell numbers during forelimb bud outgrowth[65] was mitigated by sampling identical cell numbers per developmental stage. In nuclei, tagmentation and addition of adapters were done for 30 min at 37 °C using the TN5 fusion protein from the Illumina Nextera® DNA Library kit. After Qiagen column purification, the tagmented DNA was amplified by PCR using the standard Illumina barcoded primer KAPA HiFi HotStart ReadyMix (Roche) with 13 cycles of amplification. Adapter dimers were removed using AMPure XP-beads following manufacturer instructions (Beckman Coulter). Purified libraries were pooled equimolar concentrations and sequenced using the Illumina NextSeq 500 PE 40 bases set-up.

**RNA-seq library construction**. Pairs of mouse forelimb buds at E9.75 (28–30 somites), E10.5 (34–36 somites), and E11.5 (43–45 somites) and chicken wing buds at HH20, HH22, and HH24 were dissected in ice-cold PBS and transferred to RNAlater (Sigma), stored at 4 °C overnight and then transferred to −20 °C. Tissue homogenization in RLT buffer (Qiagen) was done by sheering the sample with a 25 G needle 20 times. Total RNA was extracted using the RNeasy micro kit (Qiagen) including an on-column DNAse digestion (Qiagen) step to minimize the genomic DNA contamination. RNA integrity was checked on an

Agilent Fragment Analyser and the concentration was determined using the QubitTM RNA HS Assay kit (Ref Q32855). About 500 ng of total RNA was used to prepare RNA-seq libraries with NEB nondirectional NEBNext® Ultra™ II RNA Library Prep kit by applying the polyA+ mRNA workflow and PCR amplification for nine cycles. Barcoded RNA-seq libraries were pooled at equimolar concentrations and sequenced as single-end using Illumina NextSeq 500.

**Whole-mount RNA in situ hybridization in mouse and chicken embryos**. Standard whole-mount in situ protocols were followed for mouse[19] and chicken embryos[66]. Briefly, embryos are fixed in 4% paraformaldehyde (PFA) in PBS at 4 °C overnight, dehydrated into 100% methanol, and stored at −20 °C until further use. Following rehydration, embryos are bleached in 6% hydrogen peroxide (15 min) and then digested with 5–10 μg/ml proteinase K (10–15 min depending on the embryonic stage). Following pre-hybridization at 65 °C (≥3 h), embryos are incubated overnight at 70 °C in hybridization solution with 0.2–1 μg/ml heat-denatured antisense riboprobes to detect the transcripts of interest. The next day, embryos are extensively washed and non-hybridized riboprobe digested by 20 μg/ml RNase for 45 min at 37 °C. Subsequently, the RNA-riboprobe hybrids are revealed using the alkaline phosphatase/BMP Purple detection system (Roche-11442074001). The detection is stopped when the signal is strong but has not reached complete saturation. For comparative analysis of different stages of embryos of the same species, visualization is done for the same time. For cross-species analysis, visualization times may need to be adjusted in a species-specific manner. The results of whole-mount in situ hybridization analyses are qualitative but well suited to detect spatial changes.

**Generation of lacZ transgenic mouse founder embryos**. Mouse and chicken genomic DNA were used for PCR to amplify candidate mouse enhancers and CARs. Specific primers were designed using the Primer 3v0.4.0 web-based interface (http://bioinfo.ut.ee/primer3-0.4.0). The amplified candidate enhancer regions were inserted into the Hsp68-lacZ reporter plasmid[67] using the Gibson cloning system (New England Biolabs). Transgenic mouse founder embryos were generated by pronuclear injection at the CTM of the University of Basel. Founder embryos were isolated in ice-cold PBS around E11.5 and fixed in 1% formaldehyde, 0.2% glutaraldehyde, 0.02% NP40, 0.01% sodium deoxycholate in 1xPBS for 20–30 min at 4 °C. Subsequently, embryos were washed three times in 1xPBS for 5 min at room temperature. They were then incubated in the dark at 37 °C in a solution containing 1 mg/mL X-Gal, 0.25 mM $K_3Fe(CN)_6$, 0.25 mM $K_4Fe(CN)_6$, 0.01% NP40, 0.4 mM $MgCl_2$, and 1% sodium deoxycholate to detect ß-galactosidase activity, which colors expressing cells in blue (lacZ activity detection)[68]. The color reaction was monitored for ~6–7 h, which corresponds to the end of the exponential staining phase. Embryos with no obvious lacZ staining were left to develop overnight to detect possible weak activity. To stop the reaction, embryos were washed three times in 1x PBS for 5 min each at room temperature. An enhancer is considered robust when the expression domain of the lacZ reporter was reproducible in at least three independent founder embryos. Representative founder embryos with lacZ activities in fore- and hindlimb buds are shown for mouse enhancers and chicken CARs in Fig. 7 and Supplementary Fig. 10.

**Genome-wide analysis of accessibility landscape using ATAC-seq**. The quality of the raw paired-end sequencing reads in both species was determined by FASTQC v0.11.4[69]. Nextera primer sequences were removed with the Trim galore v0.6.2 wrapper tool for Cutadapt[70] using the --nextera --gzip --paired settings. Subsequent read alignments and post-processing was performed following ENCODE data processing standards (https://www.encodeproject.org/data-standards/). Briefly, trimmed reads were aligned with either the GRCm38/mm10 (mouse) or Gallus_gallus-5.0/galGal5 (chicken) genome build using Bowtie2 v2.2.9[71] with the -t -p 4 -X 2000 --mm -q --phred33-quals settings. Removal of PCR duplicates and generation of fragment size statistics were performed using the Picard v2.8.0[72]. SAMtools command utilities were used to remove mitochondrial and low-quality reads[73]. Enriched regions (peaks) of accessible chromatin were detected using MACS2 v2.1.1[74] with the --nomodel --shift -75 --extsize 150 -B --SPMR --keep-dup all --call-summits settings. Consistency between biological replicates was checked by Irreproducible Discovery Rate[75] and only reproducible peaks were considered for downstream analysis. A reference peak set required to identify DACs was created by merging the outer boundaries of overlapping peaks of individual stages using BEDTools v2.26.0[76]. To eliminate composition biases between libraries, trimmed mean of M-values (TMM) normalized counts were used in the generalized linear model (GLM) framework of edgeR[77] to identify the significant DACs between developmental stages. For pairwise comparisons, peaks with at least linear absolute fold change (FC) ≥ 1.5 and adjusted p values ≤ 0.05 were considered as significant DACs. Annotation of peaks in genomic regions was done using the R package ChIPseeker v1.18.0[78].

**Quantification of gene expression and differential gene expression analysis**. The quality of raw single-end sequencing reads (85 bp) was assessed using FASTQC v0.11.4[69] and the Illumina adapter sequences were trimmed using Trim Galore v0.4.1 wrapper tool for Cutadapt[70]. High-quality sequencing reads were aligned to either mouse (mm10) or chicken (galGal5) reference genomes using

STAR v2.5.2 aligner[79] with --twopassMode Basic and --quantMode TranscriptomeSAM settings. After alignment, transcripts and gene-wise counts were computed using rsem-calculate-expression utility of RSEM v1.3.0[80]. The reference gene annotation for mouse and chicken in GTF format was obtained from ENSEMBL (release 91). Prior to the identification of DEGs, all small-ncRNAs (snoRNA, miRNA, miscRNA, scRNA, and scaRNA) were filtered out. To further reduce noise and get robust outcomes, genes with the expression as counts per million reads mapped (CPM) ≥ 1 per replicate (n = 3) of the developmental stage were considered for downstream analysis. Consistency between replicates was assessed by hierarchical clustering and principal component analysis (PCA) using the hclust and prcomp R functions, respectively. Raw counts were normalized using TMM and DEGs were identified using a GLM framework with edgeR[77]. Significant DEGs were required to demonstrate a linear absolute FC cutoff of ≥1.5 and an adjusted p value ≤ 0.05.

**DAC and DEG module analysis**. As described by Gray and colleagues[38], k-means clustering was employed on significantly enriched DACs and DEGs to identify modules of chromatin accessibility (DAC modules) and gene expression (DEG modules). First, peak accessibility values of each peak were scaled between 0 and 1 using min-max normalization as follows: the minimum value across all three stages was subtracted, and then values were divided by the maximum. The same scaling was performed on gene expression data for DEGs. Assuming two states of chromatin accessibility i.e., open/close and given three stages, $2^3$ combinations are possible for a peak's trajectory. Similarly, two states of gene expression (expressed/not expressed) across three stages result in eight combinations for a gene expression trajectory. As two combinations (0,0,0) and (1,1,1) correspond to no differential temporal trajectory, the remaining six combinations were taken as seeds for k-means clustering to identify the convergent set of DAC and DEG modules. This permitted use of the same starting conditions for both ATAC-seq peak and gene expression clustering. Furthermore, the robustness and separation distance between the resulting clusters were checked using silhouette analysis[81]. This measure has a range of [−1,1] and a positive silhouette score such that it highlights that the samples are away from the decision boundary between two neighboring modules[81]. Using these robust DAC and DEG modules, the significance of peak-gene module associations was tested. By calculating the frequency with which peaks in each DAC module were positionally associated with the genes in each DEG module, significant enrichments or depletions were computed using Fisher's exact test. The Benjamini and Hochberg (BH) correction was used to account for multiple comparisons.

**Generation of the genome-wide ATAC-seq and RNA-seq signals**. Normalized genome-wide profiles of ATAC-seq and RNA-seq samples were generated using DeepTools[82]. The Binary Alignment Map (BAM) alignments were converted to bigwig format using the bam-Coverage utility of the DeepTools suite to calculate the normalized signals in defined windows (50 bp default size) as fragments per kilobase million (FPKM) for paired-end data and reads per kilobase million (RPKM) for single-end sequence data. The normalized signal coverage tracks with replicates overlaying each other were generated and visualized at UCSC Genome Browser[83].

**Orthologous genes and developmental stage correspondence**. A set of orthologous genes between mouse and chicken was retrieved using the BioMart application from ENSEMBL (https://www.ensembl.org). To obtain a high-confident set, orthologous genes from both species with Protein stable ID and ortholog_one2one homology type were considered for the stage correspondence between mouse and chicken limb buds. The transcript levels of orthologous genes were measured in transcripts per million (TPM) and the Spearman's correlation coefficient (ρ) was computed to deduce the transcriptome similarity between developmental stages of mouse and chicken limb buds.

**Functional annotation enrichment analysis**. For each species, the functional annotations correspond to DEGs of DEG modules were queried using the g:Profiler[84] python API against all genes of the respective organism as background. The Benjamini–Hochberg (BH) was applied for multiple testing corrections with the significance set at an FDR of 0.05.

**Motif enrichment analysis**. To evaluate DACs for binding motif enrichment, curated position weight matrices (PWMs) of nonredundant core vertebrate TF binding profiles from the JASPAR 2018 database[39] were used, while the PWMs for the GLI3 and HAND2 binding motifs were obtained from CIS-BP[85]. As described[38], a background set for each foreground module was first generated such that peaks in the foreground and background modules do not share overlapping accessibility patterns. The fasta sequences of peaks in foreground and background modules were extracted using BEDTools v2.26.0. Files containing fasta sequences were submitted to analysis of motif enrichment (AME) v5.0.4[86] to compute enrichment of sequence motifs within DAC modules against dissimilar background sets using the --method fisher --scoring avg settings. Similarly, AME was also performed with foreground and background peak sets reversed to identify motifs underrepresented or depleted in the DAC modules.

**Genome-wide prediction of TFBS motifs**. Using the PWMs of TFs binding profiles, the genomic locations of TF motifs ($p < 1e-4$) were predicted in reference peak set of the mouse (mm10) and chicken (galGal5) using FIMO utility as part of the MEME suite (v5.0.4).

**TF footprint analysis**. Using stage-specific deep ATAC-seq, computational footprints of select TF motifs were inferred using the rgt-hint utility of the application HINT part of the Regulatory Genomics Toolbox[87]. TFs bound to DNA prevent Tn5 transposase from cleavage in an otherwise nucleosome-free region, which leads to drops in occupancy within peak regions of high coverage resulting in a typical TF footprint. Using BAM-formatted mapped reads and differential accessible sites, the genomic locations of footprints for select TF motifs were detected using the rgt-hint commands in footprinting mode with *--atac-seq --paired-end --organism* parameters. These footprints were checked for the presence of binding motifs of TFs by overlapping them with the motif location predictions given by FIMO in the peak sets. Finally, bias-corrected normalized signal profiles for each of the motifs were obtained using the *--organism --bc --window-size 200* settings. Motif-centered aggregated footprint profiles with a moving average (window size = 5 bp) were plotted for each developmental stage using custom Python codes. Motif LOGOS were downloaded from JASPAR. For motifs other than JASPAR, web LOGOS were generated using the seqLogo R library[88].

**Identification of candidate HAND2 and GLI3 target genes**. Using a subset of key genes and TFs known to function during limb bud patterning and outgrowth (Supplementary Dataset 6), HAND2 and GLI3 target genes were identified for mouse forelimb and chicken wing buds. Similar to Gray et al., 2017[38], our analysis is based on the assumption that if TF-X (X = GLI3/HAND2) shows binding evidence within the putative *cis*-regulatory region associated with gene Y, then Y is a target gene regulated by TF-X. To establish these potential interactions (edges) between source TF and target genes, we checked for the presence of source TFBS within chromatin accessible regions associated to target genes, thereby establishing a connection from source TF to a target gene. Using the motif location predictions given by FIMO in our peak datasets, we identified high-confident motifs of source TFs in peaks positionally associated with the set of potential targets[38]. The resulting diagram with nodes and connections was plotted using the Cytoscape v.3.7.2[89] python API. The edge width of the connections in the network diagram corresponds to the number of TF binding motifs establishing a connection between the source TF (GLI3/HAND2) and the target gene. Each target gene was assigned the closest DEG module by calculating the Pearson's correlation coefficient between average gene expression and the cluster centroids of each DEG module. For each species, the edges are colored to show either shared interactions mediated by both GLI3 and HAND2 (gray) or TF-specific (GLI3 or HAND2) interactions (magenta) with a particular target gene. For cross-species comparison of conserved and species-specific interactions between mouse and chicken, the regulatory networks were compared by computing the similarity score[6]. Briefly, considering both nodes and their connections, the similarity score can be defined as:

$$S = 1 - D \qquad (1)$$

where *D* represents differences between mouse and chicken limb buds and can be defined as:

$$D = \frac{1}{2}\left(\frac{N}{M} + \frac{I}{2J}\right) \qquad (2)$$

where, *N* represents the number of different connections, *M* corresponds to the number of all connections, *I* represents the number of different target genes, and *J* corresponds to the number of all target genes.

**Computational identification of CARs**. CARs were identified using a strategy similar to Booker et al., 2016[12]. Briefly, a comprehensive set of previously characterized mouse limb bud enhancers active between E9.5 and E12.0 was assembled from VISTA[45] and CAGE[46] databases in combination with Capture-C[47] analysis of mouse limb buds. Prior to identifying vertebrate conserved elements, the chicken genome was excluded from the 60-way vertebrate multiple sequence alignments with the mouse (mm10) as reference genome (UCSC Genome Browser) to include those elements that might possess high rates of nucleotide differences in chicken, yet otherwise conserved in vertebrates. The resulting genome-wide alignments were used as input to PhastCons[90] to identify conserved noncoding regions. Next, short conserved regions in close proximity were iteratively merged and the merged regions were intersected with the mouse limb bud enhancer dataset. Regions with >50% of the sequence missing in the chicken genome were eliminated. These resulting candidate regions that overlap with mouse limb bud enhancers were tested for accelerated nucleotide substitutions in the chicken genome using PhyloP function of package PHAST[91] with the *--branch* option. The false discovery rate (FDR) was computed using the Benjamini–Hochberg method and regions with an FDR < 5% were considered as candidate CARs that displayed significantly accelerated nucleotide substitutions in the chicken genome. Each CAR was assigned the closest gene using GREAT[92]. In addition, genes expressed/functioning in limb buds that reside within 1MB of each CAR were identified. Further, the species-specific DACs were overlapped with CARs (≤2 kb window) to identify different CAR-DAC

categories. Moreover, the CAR coordinates were intersected with putative chicken embryonic and/or limb enhancers and ASHCEs[48] using BEDTools v2.26.0[76]. The overrepresentation of predicted enhancer classes in CARs was tested using GAT[93].

**Available datasets used in this study**. The information on the gene markers used for our study has part of the single-cell transcriptome analysis during mouse organogenesis[94]. The information of a set of genes whose mutagenesis, loss-of-function, and gain-of-function in mouse models results in limb congenital malformations was obtained from the Mouse Genome Informatics database (www.informatics.jax.org/). TF family information for mouse and chicken genes were obtained from Animal TFDB v3.0[95]. The peaks of histone modifications for mouse limb developmental stages were obtained from mouse ENCODE (https://www.encodeproject.org/data/). The genomic coordinates of ASHCEs and the predicted putative enhancers of chicken embryos and limb buds have been previously published[48].

**Reporting Summary**. Further information on research design is available in the Nature Research Reporting Summary linked to this article.

## Data availability

The newly sequenced genome-wide RNA-seq and ATAC-seq datasets of mouse and chicken limb buds used in the study have been deposited to Gene Expression Omnibus (GEO) public repository. The RNA-seq datasets are available under the accession code GSE164737 and ATAC-seq datasets at GSE164736. Additional processed datasets are included as Supplementary Datasets. Source data are provided with this paper.

## Code availability

The repository containing the custom scripts used for analyses is available at Zenodo (https://doi.org/10.5281/zenodo.5172892).

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

## Acknowledgements

We thank A. Morabito and T. Oberholzer for expert assistance with RNA in situ hybridization analysis of mouse and chicken embryos. We thank E. Terszowska and A. Offinger for excellent mouse care and the Quantitative Genomics Facility of the University and ETH Basel for sequencing the genome-wide datasets generated for this study. All the computational analysis was performed using the Scientific Computing Center at the University of Basel (https://scicore.unibas.ch). P. Pelczar and his team at the University of Basel Center for Transgenic Models are acknowledged for generating all transgenic mouse founder embryos for the comparative analysis of CAR activities. We are indebted to M. Ros, N. Vargesson, and P. Tschopp for providing plasmids encoding chicken and mouse riboprobes. This research was supported by the ERC Advanced Grant INTEGRAL ERC-2015-AdG; Project ID 695032 and core funding by the University of Basel.

## Author contributions

S.J. conceived the study and performed data curation, coding, visualization, and computational analysis. J.M. generated and analysed the *lacZ* reporter constructs with assistance by J.S. J.S. generated the genome-wide datasets used for this study. O.R. and J.M. performed the whole-mount in situ hybridization analysis in mouse and chicken limb buds. S.J., A.Z., and R.Z. discussed the results and their interpretation, proposed the comparative and transgenic analysis in embryos, and together with S.J. prepared all figures and tables and wrote the manuscript. All authors gave input into the manuscript before submission.

## Competing interests

The authors declare no competing interests.
