## [Peer Review File · Nature Communications]

Conserved and species-specific chromatin remodelling and regulatory dynamics during mouse and chicken limb bud developmentReviewers' Comments:

Reviewer #1:

Remarks to the Author:

This manuscript by Shalu Jhanwar et al. addresses the temporal dynamics of chromatin accessibility and its relationship with gene expression dynamics during limb and wing development in the mouse and chicken models, respectively. The study combines ATACseq and RNAseq data generated from three equivalent stages of limb and wing development (E9.75, E10.5 and E11.5 for mouse embryos; HH20, 22 and 24 for chicken embryos), with extensive bio-informatic analyses to characterize limb and wing development. These analyses thus identify sets of genes showing similar temporal expression dynamics (grouped in six modules, ie DEG modules) as well as sets of chromatin regions sharing similar dynamics of accessibility (also grouped in 6 modules, ie DAC modules) and reveal (1) the synchrony between chromatin accessibility and gene expression in the mouse forelimb buds in contrast with the chicken wing buds. They also reveal (2) similarities and differences in gene expression dynamics between chicken and mouse appendage development. (3) Combined with in silico predictions, the chromatin accessibility and gene expression data support stage-specific and dynamic transcription factor binding on accessible chromatin regions to control target genes. Next, (4) in situ hybridization and reporter transgenes also identify conserved or markedly species-specific spatial patterns of gene expression. Finally, (5) comparative genomics allowed identifying chicken sequences showing accelerated evolution contributing to chicken-specific cis-regulation in limb buds. This work is elegant, robust, based on a wealth of data and very well-presented. The data presented in this manuscript are of importance for the field of limb development, since it identifies, in a holistic approach, sets of genes and sets of chromatin domains showing correlated (or not) temporal dynamics during limb development. They also offer an evo-devo perspective while comparing the mouse and chicken models showing shared but also divergent molecular patterns (chromatin openness, differentially expressed genes). In addition to these important data, this study is also very significant for its methodological rationale. The authors demonstrate that integrating ATACseq, RNAseq and in silico analyses provides a unique way to establish candidate transcription factor/target gene networks. Its significance to the field of developmental biology, its evo-devo conclusions and its original methodological design make it appropriate for publication in a wide-audience journal like Nat Comm.

Only very few minor points need to be addressed to improve the manuscript.

A few abbreviations should be defined upon their first occurrence in the body of the text like AER, CPM, PCA, BAM; as well as in figure legends like AME (legend to Figure 6).

On Figure 8 as well as on Figure S8, the ATACseq profiles are referred in the legend to be presented in "magenta". But looking at the figures these ATACseq signals seem to be presented with distinct shades of color from light "pink" to magenta. Why distinct color shades were used in presenting these ATACseq profiles should be clarified.

On page 15, while referring to MEIS2, the authors indicate this to be a bHLH protein, which is not correct. MEIS proteins are TALE homeodomain proteins.

In the legend to Figure S4.g, there is a typo: "DASs" should appear as "DACs".

Reviewer #2:

Remarks to the Author:

This manuscript by Jhanwar et al. reports the comprehensive genomics data and in-depth computational analysis intersecting RNA-seq and ATAC-seq data from mouse and chicken limb buds. They identified the differentially accessible chromatin regions (DACs) and differentially expressed genes (DEGs) during limb development. Comparative analysis of these genomics data revealed the differentially and similarly expressed genes between the two species and cis-regulatory network of GLI3 and HAND2 with candidate target genes in limb buds. Some target genes predicted in this model are verified in the previous articles, and this *in silico* model looks reliable. They also identified the chicken accelerated regions (CARs) from the aspect of DNA sequence evolution and showed

differential enhancer activity between mice and chickens.

Overall, *in silico* analysis of genomics data (RNA expression, chromatin state, and DNA accelerations) are intensive and provides great insight into species-specific gene regulation. This manuscript should be considered for publication in Nature communications.

Major concerns.

1. I am interested in if CARs contribute to the species-specific differential expression. Do authors see the association of CARs and DACs/DEGs?

Since most CARs' enhancer activity are weak compared to mice, these regions are likely to overlap with DAGs. The genome-wide analysis of CARs and DACs would be great to understand the role of DNA sequence accelerations in species-specific gene regulation.

2. If it is possible, please show the mouse and chicken whole-mount *in situ* hybridization data for CARs' target genes shown in Fig8.

3. Avian-specific highly conserved elements and enhancers have been identified before (Seki et al. 2017, Nature Commun. DOI: 10.1038/ncomms14229). The association of these elements and chicken DACs/CARs should be analyzed. This analysis would help to justify chicken-specific DACs data.

Minor concerns.

1. Authors claimed that ChIP does not work in the chicken, but several groups have done CTCF, histone modifications, and TF ChIP-seq (Yakushiji-Kaminatsui et al. 2018 Plos Biol, Seki et al. 2017, Nature Commun and Ref 57, 58 in the manuscript). I am not sure if this is a proper justification for this research. (such as for Page 5 line1)

2. Please make clear the meaning of abbreviation of ACR (FigS2g) and GRN (in the main text).

Reviewer #3:

Remarks to the Author:

The paper by Jhanwar et al characterises and compares three critical stages of limb bud development in the two main model organisms used in the field (mouse and chicken). Using RNA seq to determine gene expression and ATAC to assess chromatin accessibility, they produce comprehensive resources which will be of great interest and benefit to the community. Much of the data presented is the careful and robust validation of these libraries.

Perhaps if I were a bioinformatician it would be clearer to me, but could the authors explain somewhere for the lay-er reader the distinction between, for example, 9.75lo and 11.5Hi DEGS and vis versa as both classes seem to share an expression trajectory meaning that 55% of DEGs start low and gain expression with time. The distinction seems especially important as the converse is often the second largest category in Figure S5.

In the experiments to compare the spatial patterns of expression between mouse and chicken seem somewhat of an outlier in the rest of the data. While I accept that may not be possible to directly compare the qualitative distribution with quantitative levels, a number of those genes appear so discordant in even trends that I wonder what this adds to their story and if this data might not be better omitted and instead some of it used in the discussion of the CARs (see below)? At the very least, it would be helpful to have the genes in 5A annotated as to which DEG module they represent.

The identification of sequences with accelerated sequence changes in chicken (CARs) is interesting but given that they were selected on the basis of sequences divergence from the mouse enhancers, it seems unsurprising that they drive different expression in LacZ transgenics. Could this work be extended to more closely compare the activity of the species-specific enhancers to the endogenous expression patterns? A few of those genes examined (Prrx2, Smad7, Id1 and Msx1) are currently omitted from figure 5.

Response to Reviewers Comments and Suggestions

We are grateful to the three reviewers for their positive evaluation of our study and the constructive comments and suggestions. We have performed all proposed additional bioinformatics and experimental analysis and revised the manuscript taking into account all comments and suggestions. In our opinion, the revised manuscript is significantly improved and more comprehensive.

Reviewer #1

This manuscript by Shalu Jhanwar et al. addresses the temporal dynamics of chromatin accessibility and its relationship with gene expression dynamics during limb and wing development in the mouse and chicken models, respectively. The study combines ATACseq and RNAseq data generated from three equivalent stages of limb and wing development (E9.75, E10.5 and E11.5 for mouse embryos; HH20, 22 and 24 for chicken embryos), with extensive bio-informatic analyses to characterize limb and wing development. These analyses thus identify sets of genes showing similar temporal expression dynamics (grouped in six modules, ie DEG modules) as well as sets of chromatin regions sharing similar dynamics of accessibility (also grouped in 6 modules, ie DAC modules) and reveal (1) the synchrony between chromatin accessibility and gene expression in the mouse forelimb buds in contrast with the chicken wing buds. They also reveal (2) similarities and differences in gene expression dynamics between chicken and mouse appendage development. (3) Combined with in silico predictions, the chromatin accessibility and gene expression data support stage-specific and dynamic transcription factor binding on accessible chromatin regions to control target genes. Next, (4) in situ hybridization and reporter transgenes also identify conserved or markedly species-specific spatial patterns of gene expression. Finally, (5) comparative genomics allowed identifying chicken sequences showing accelerated evolution contributing to chicken-specific cis-regulation in limb buds. This work is elegant, robust, based on a wealth of data and very well-presented. The data presented in this manuscript are of importance for the field of limb development, since it identifies, in a holistic approach, sets of genes and sets of chromatin domains showing correlated (or not) temporal dynamics during limb development. They also offer an evo-devo perspective while comparing the mouse and chicken models showing shared but also divergent molecular patterns (chromatin openness, differentially expressed genes). In addition to these important data, this study is also very significant for its methodological rationale. The authors demonstrate that integrating ATACseq, RNAseq and in silico analyses provides a unique way to establish candidate transcription factor/target gene networks. Its significance to the field of developmental biology, its evo-devo conclusions and its original methodological design make it appropriate for publication in a wide-audience journal like Nat Comm.

Only very few minor points need to be addressed to improve the manuscript. A few abbreviations should be defined upon their first occurrence in the body of the text like AER, CPM, PCA, BAM; as well as in figure legends like AME (legend to Figure 6).

As suggested, we have defined the abbreviations upon their first occurrence in the manuscript. In addition, we added a list of all abbreviations to the Methods section.

On Figure 8 as well as on Figure S8, the ATACseq profiles are referred in the legend to be presented in “magenta”. But looking at the figures these ATACseq signals seem to be presented with distinct shades of color from light “pink” to magenta. Why distinct color shades were used in presenting these ATACseq profiles should be clarified.

The distinct colour shades were used to indicate the two ATAC-seq replicates and their overlap. In the figure legends we now describe the different color shades used for ATAC-seq profiles.

On page 15, while referring to MEIS2, the authors indicate this to be a bHLH protein, which is not correct. MEIS proteins are TALE homeodomain proteins. In the legend to Figure S4.g, there is a typo: “DASs” should appear as “DACs”.

We thank the reviewer for spotting these errors that have been corrected in the revised manuscript.

Reviewer #2

This manuscript by Jhanwar et al. reports the comprehensive genomics data and in-depth computational analysis intersecting RNA-seq and ATAC-seq data from mouse and chicken limb buds. They identified the differentially accessible chromatin regions (DACs) and differentially expressed genes (DEGs) during limb development. Comparative analysis of these genomics data revealed the differentially and similarly expressed genes between the two species and cis-regulatory network of GLI3 and HAND2 with candidate target genes in limb buds. Some target genes predicted in this model are verified in the previous articles, and this in silico model looks reliable. They also identified the chicken accelerated regions (CARs) from the aspect of DNA sequence evolution and showed differential enhancer activity between mice and chickens. Overall, in silico analysis of genomics data (RNA expression, chromatin state, and DNA accelerations) are intensive and provides great insight into species-specific gene regulation. This manuscript should be considered for publication in Nature communications.

Major concerns.

- 1. I am interested in if CARs contribute to the species-specific differential expression. Do authors see the association of CARs and DACs/DEGs? Since most CARs' enhancer activity are weak compared to mice, these regions are likely to overlap with DAGs. The genome-wide analysis of CARs and DACs would be great to understand the role of DNA sequence accelerations in species-specific gene regulation.*

As suggested, we have intersected the CARs with the species-specific DACs and DEGs that we have identified (Supplementary Fig. 9c-e, Supplementary Table 12). We found that 39.5% of CARs were differentially accessible in chicken wing buds (CAR-gg-DAC and CAR-mm/gg-DAC) while the others were differentially accessible either in mouse (21%, CAR-mm-DAC) or none of the species (39.5%, CAR-no-DAC, Supplementary Fig.9c-e). Furthermore, 50.95% of the CARs have at least one putative target gene among its assigned targets that was DEGs in

either or both species (Supplementary Fig. 9f-h, Supplementary Table 12). GO analysis of the CAR-associated DEGs points to functions in limb and embryonic development/morphogenesis (Supplementary Fig. 9f). We describe these results on page 19.

If it is possible, please show the mouse and chicken whole-mount in situ hybridization data for CAR's target genes shown in Fig 8.

We now include mouse and chicken whole-mount *in situ* hybridization data for the target genes of all CARs analyzed (Fig.7 and Supplementary Fig. 10). This analysis reveals the spatial changes in the expression of *Tbx2* and *Smad7* in mouse and chicken limb buds, which overlap the differential spatial *lacZ* activities of the CAR and its mouse orthologue (Fig. 7a-b). The differential activities of the other CARs and their mouse cognate enhancers are not paralleled by obvious spatial changes in the expression of the associated gene at the orthologous limb bud stages in both species. We describe these results on pages 19-21 and also discuss these findings in the context of accelerated regions in other species (pages 24-25).

2. Avian-specific highly conserved elements and enhancers have been identified before (Seki et al. 2017, Nature Commun. DOI: 10.1038/ncomms14229). The association of these elements and chicken DACs/CARs should be analyzed. This analysis would help to justify chicken-specific DACs data.

As suggested, we have analyzed the association of avian-specific highly conserved elements (ASHCE) and putative enhancers (Seki et al., 2017) with our chicken CARs/DACs datasets. Our unbiased analysis showed that more than 96% of all CARs are not part of avian-specific ASHCEs, suggesting that the identified CARs are indeed chicken-specific (Supplementary Table 8). In addition, 12.82% of the chicken DACs overlap ASHCEs (Supplementary Table 10).

An additional interesting finding was that the majority of all CARs (78%) and DACs (90%) overlap putative chicken embryonic and/or limb enhancers that have been previously predicted by Seki et al. (2017) using combinatorial patterns of epigenetic signatures, which corroborates their potential roles in *cis*-regulation of gene expression (Supplementary Table 8,10, Supplementary Fig. 9b). In addition, we performed an over-representation analysis of different activity states of the predicted enhancers (*strong*, *weak*, *poised*, Seki et al., 2017) in CARs (Supplementary Table 8-9) and chicken DACs (Supplementary Table 10-11), which showed that *strong* enhancers are over-represented in these two datasets. This additional analysis is described on pages 18-19 of the revised results section.

Minor concerns.

1. Authors claimed that ChIP does not work in the chicken, but several groups have done CTCF, histone modifications, and TF ChIP-seq (Yakushiji-Kaminatsui et al. 2018 Plos Biol, Seki et al. 2017, Nature Commun and Ref 57, 58 in the manuscript). I am not sure if this is a proper justification for this research. (such as for Page 5 line1).

We agree with the reviewer that this generalized statement is not correct and in fact it is not needed to justify the research. We have removed it from the revised manuscript.

2. *Please make clear the meaning of abbreviation of ACR (FigS2g) and GRN (in the main text).*

ACR is a typo in FigS2g, it should read DAC (corrected in Fig. S2g) and the abbreviation of GRN (gene regulatory network) has been defined in the text.

Reviewer #3 (Remarks to the Author)

The paper by Jhanwar et al characterises and compares three critical stages of limb bud development in the two main model organisms used in the field (mouse and chicken). Using RNA seq to determine gene expression and ATAC to assess chromatin accessibility, they produce comprehensive resources which will be of great interest and benefit to the community. Much of the data presented is the careful and robust validation of these libraries. Perhaps if I were a bioinformatician it would be clearer to me, but could the authors explain somewhere for the lay-er reader the distinction between, for example, 9.75^{lo} and 11.5^{Hi} DEGS and vis versa as both classes seem to share an expression trajectory meaning that 55% of DEGs start low and gain expression with time. The distinction seems especially important as the converse is often the second largest category in Figure S5.

In the revised results section we now include a detailed description that explains why the kinetics of these DEG modules differ significantly (page 7):

“Although genes belonging to E9.75^{hi} and E11.5^{lo} DEG modules show the highest expression at E9.75 and their expression decreases during limb bud outgrowth, these two modules are distinct due to their significantly different expression levels at the transition point (E10.5, top and bottom panels in Fig. 1d; box plots in Supplementary Fig. 1c). In particular, the decrease in gene expression between E9.75 and E10.5 is much larger for the E9.75^{hi} than the E11.5^{lo} DEG module. The converse is true for genes belonging to the E9.75^{lo} and E11.5^{hi} DEG modules. Overall, all DEG modules display distinct expression kinetics in developing mouse forelimb buds.”

In the experiments to compare the spatial patterns of expression between mouse and chicken seem somewhat of an outlier in the rest of the data. While I accept that may not be possible to directly compare the qualitative distribution with quantitative levels, a number of those genes appear so discordant in even trends that I wonder what this adds to their story and if this data might not be better omitted and instead some of it used in the discussion of the CARs (see below)? At the very least, it would be helpful to have the genes in 5A annotated as to which DEG module they represent.

We agree with this reviewer and these data are now shown in Supplementary Fig. 6. To avoid confusion due to directly comparing the qualitative distribution measured by *in situ* with quantitative levels given by RNA-seq, we have followed this reviewer’s suggestions and: 1. removed panel 5A. 2. indicated the DEG modules for all genes

whose spatio-temporal expression is analysed for both mouse and chicken limb buds in the figure legend. The results of this analysis are described with a focus on the main findings on page 14.

The identification of sequences with accelerated sequence changes in chicken (CARs) is interesting but given that they were selected on the basis of sequences divergence from the mouse enhancers, it seems unsurprising that they drive different expression in LacZ transgenics. Could this work be extended to more closely compare the activity of the species-specific enhancers to the endogenous expression patterns? A few of those genes examined (Prrx2, Smad7, Id1 and Msx1) are currently omitted from figure 5.

This is indeed an important point and we now include mouse and chicken whole-mount *in situ* hybridization data for the target genes of all CARs analyzed (Fig.7 and Supplementary Fig. 10). This analysis reveals the spatial changes in the expression of *Tbx2* and *Smad7* in mouse and chicken limb buds, which overlap the differential spatial *lacZ* activities of the CAR and its mouse orthologue (Fig. 7a-b). The differential activities of the other CARs and their mouse cognate enhancers are not paralleled by obvious spatial changes in the expression of the associated gene at the orthologous limb bud stages in both species. We describe these results on pages 19-21 and also discuss these findings in the context of accelerated regions in other species (pages 24-25).

Reviewers' Comments:

Reviewer #1:

Remarks to the Author:

Shalu Jhanwar et al. appropriately took reviewer's comments into consideration and modified their manuscript accordingly. The manuscript in its revised form is therefore suitable for publication.

Reviewer #2:

Remarks to the Author:

The authors have addressed all comments made by the reviewers and I thus support publication of this manuscript in Nature Communications.

Reviewer #3:

Remarks to the Author:

I like the revisions the authors have made to the manuscript. The section on the CARs in particular has been much improved.

Its a lovely paper which will be of great interest to the community.

I'm happy to recommend publication.

Laura Lettice